

# Footprint-scale cloud type mixtures and their impacts on Atmospheric Infrared Sounder cloud property retrievals

Alexandre Guillaume[1], Brian H. Kahn[1], Eric J. Fetzer[1], Qing Yue[1], Gerald J. Manipon[1], Brian D. Wilson[1], Hook Hua[1]

[1] Jet Propulsion Laboratory, California Institute of Technology, Pasadena, 91109, USA

*Correspondence to*: Alexandre Guillaume (Alexandre.Guillaume@jpl.nasa.gov)

**Abstract.** A method is described to classify cloud mixtures of cloud top types, termed *cloud scenes,* using cloud type classification derived from the CloudSat radar (2B-CLDCLASS). The scale dependence of those cloud scenes is studied. For spatial scales near 45 km, only 16 out of 256 possible cloud scenes account for 90% of all observations and contain either one,

two, or three cloud types. The number of possible cloud scenes is shown to depend on spatial scale with a maximum number of 194 out of 256 possible scenes at a scale of 105 km and fewer cloud scenes at smaller and larger scales. The cloud scenes are used to assess the characteristics of spatially collocated Atmospheric Infrared Sounder (AIRS) thermodynamic phase and ice cloud property retrievals within scenes of varying cloud type complexity. The likelihood of ice and liquid phase detection strongly depends on the CloudSat-identified cloud scene type collocated with the AIRS footprint. Cloud scenes primarily

consisting of cirrus, nimbostratus, altostratus and deep convection are dominated by ice phase detection, while stratocumulus, cumulus, and altocumulus are dominated by liquid and undetermined phase detection. Ice cloud particle size and optical thickness are largest for cloud scenes containing deep convection and cumulus, and are smallest for cirrus. Cloud scenes with multiple cloud types have small reductions in information content and slightly higher residuals of observed and modelled radiance compared to cloud scenes with single cloud types. These results will help advance the development of temperature,

specific humidity, and cloud property retrievals from hyperspectral infrared sounders that include cloud microphysics in forward radiative transfer models.

## 1 Introduction

There is increasing evidence of secular cloud trends at regional and global scales in both satellite observations (e.g., Norris et al. 2016) and climate general circulation model (GCM) simulations (e.g., Zelinka et al., 2013). The poleward migration of the

extratropical storm tracks (Barnes and Polvani, 2013) is coupled with systematic changes in cloud thermodynamic phase partitioning in forced $CO_2$ experiments in climate GCMs (e.g., Mitchell et al. 1989; Ceppi et al. 2016). The spread in equilibrium climate sensitivity is also tightly coupled to the temporal evolution of phase partitioning in most climate GCMs (Tan et al. 2016). Obtaining reasonable observational estimates of the small-scale cloud phase partitioning at model sub-grid scales are critical for constraining the highly uncertain Wegener–Bergeron–Findeisen time scale parameter that is crucial for



modelling mixed phase cloud and precipitation processes and (Tan and Storelvmo 2016). A new generation of probability distribution function-based parameterizations has shown promise for improving climate model simulations of cloud properties (e.g., Golaz et al. 2002) and would benefit from further exploitation of the information available in pixel-scale satellite observations. A rigorous assessment of the scale dependence of cloud types, and their mixtures, would also enhance climate

GCM evaluation and parameterization development research efforts (Bony et al. 2006).

Kahn et al. (2018) showed that Atmospheric Infrared Sounder (AIRS) observations of ice cloud optical thickness ($\tau_i$) and effective radius ($r_{ei}$) exhibit statistically significant temporal trends that are dependent on latitude and cloud type. Trends in Multi-angle Imaging SpectroRadiometer (MISR) observations of cloud texture have suggested that recent thinning of tropical

cirrus has led to increased detection of trade cumulus (Zhao et al. 2016). Using high-spatial-resolution estimates of cloud thermodynamic phase obtained from the Hyperion instrument on Earth Observing 1 (EO-1), Thompson et al. (2018) showed that phase mixtures are highly variable at scales smaller than the AIRS footprint or typical GCM grid boxes. These studies (and many others) suggest that quantification of the scale dependence of cloud type mixtures could help explain satellite observations of cloud trends.

Statistical classification methods are commonly used to define weather states or cloud types (e.g., Rossow et al., 2005; Xu et al., 2005; Sassen and Wang, 2008; Wang et al. 2016). For instance, joint histograms of cloud top pressure and optical thickness from the International Satellite Cloud Climatology Project (ISCCP; Rossow and Schiffer 1999) are useful for relating cloud types to dynamical, radiation and precipitation variability, and in evaluating climate model simulations (e.g., Klein and Jakob,

1999; Jakob and Tselioudis 2003; Rossow et al. 2005; Tselioudis et al., 2013). Weather states are typically mixtures of conventional cloud types as shown by Rossow et al. (2005) and Oreopoulos et al. (2014). Partly inspired by this methodology, we introduce the concept of *cloud scenes* that are defined to be mixtures of CloudSat cloud types (2B-CLDCLASS; Sassen and Wang 2005) that vary with horizontal scale.

As cloud scenes will be matched to coincident A-train observations or reanalysis datasets, we begin by defining cloud scenes with cloud types derived from CloudSat and observed within an AIRS/Advanced Microwave Sounding Unit (AMSU) (Chahine et al. 2006) field of regard (FOR) of roughly 45-km resolution. One AIRS/AMSU FOR within an AIRS/AMSU swath is spatially and temporally coincident with a 'curtain' of 94 GHz CloudSat radar profiles. The likelihood of observing clouds is resolution-dependent and is approximately 80–85% at the AIRS footprint scale of 15 km (Krijger et al. 2007; Kahn et al.,

2008). The clouds in AIRS/AMSU sounding FORs or AIRS footprints are more often broken or transparent, and less often uniform or opaque. Yue et al. (2013) showed that about 43% of the AIRS/AMSU FORs are mixtures of CloudSat-identified cloud types, implying that roughly half of cloudy soundings contain mixtures of cloud types.



Our purpose in this work is to quantify the scale dependence of cloud type mixtures that are then used to understand the cloud complexity within AIRS cloud phase and ice cloud property data sets. The AIRS and CloudSat data and the collocation approach are described in Section 2. To quantify cloud type distributions and their dependence on horizontal scales, the cloud scenes are first characterized at the AIRS/AMSU FOR resolution in Section 3.1, are extended to larger and smaller scales in

Section 3.2, and key results of the scale dependence are placed into context in Section 3.3. The cloud scenes are used to partition AIRS cloud property retrievals into cloud types, specifically, cloud thermodynamic phase histograms in Section 4.1, and mean values of ice cloud microphysical parameters are described in Section 4.2. A discussion, summary, and suggestions for future investigation are found in Section 5.

## 2 Data and Methodology

**2.1 CloudSat and AIRS pixel-scale matching**

The AIRS/AMSU/CloudSat matchup product described in Manipon et al. (2012) is used by Yue et al. (2013) and in this investigation. The matchup process uses a nearest neighbor approach to geolocate all CloudSat profiles within either an AIRS/AMSU FOR at 45 km spatial resolution at nadir view, or a single AIRS footprint at 15 km spatial resolution at nadir view (Kahn et al. 2008). Approximately 45 to 50 (15–17) CloudSat profiles coincide with a single AIRS/AMSU FOR (AIRS

footprint), in a swath of width 30 FOR (90 footprints). The cloud scenes are first defined at the AIRS/AMSU FOR scale and are then extended to other spatial scales. We use a two-year period of data extending from 01 July 2006 until 30 June 2008 which contains about 8 million AIRS/AMSU FORs (or 24 million AIRS footprints).

**2.2 CloudSat cloud types and their mixtures within the AIRS footprint**

There are eight CloudSat defined classes in the 2B-CLDCLASS files: cumulus (Cu), stratocumulus (Sc), stratus (St),

altocumulus (Ac), altostratus (As), nimbostratus (Ns), cirrus (Ci), and deep convective (Dc) clouds, with a ninth classification of clear sky designated no cloud (nc). Since each AIRS/AMSU FOR contains roughly 50 CloudSat profiles with 125 vertical levels each, there are $9^{50 \times 125}$ possible distinct cloud type combinations (although in practice there are fewer as many levels reside in the stratosphere) for each AIRS/AMSU FOR. This number is too high to derive a classification that could be useful, i.e. where each cloud type combination could be populated with a significant number of samples for any climatological study.

One particularly appealing way to reduce the dimensionality is to limit consideration of cloud type to cloud top only. This simplification is consistent with the capabilities of infrared sounders as the sampling of temperature and specific humidity is maximized in the atmosphere near and above cloud top, assuming the cloud is opaque and covers the entire sounder pixel area. There are $9^{50}$ possible cloud combinations defined in this manner including clear sky profiles. As comparison, there are about 324,000 AIRS soundings per day or about $10^8$ per year. Even when considering the 16 years of AIRS nominal operation, the

number of cloud type combinations $9^{50}$, or about $5 \times 10^{47}$, is many orders of magnitude greater than the number of AIRS/AMSU





FORs available, making impossible a statistically significant sampling of all combinations. This necessitates further assumptions to define a practical yet meaningful set of cloud scenes.

Two additional simplifications are made here: variations in the count of each CloudSat cloud type is not considered, and the observation sequence of successive cloud types is disregarded. These two simplifications are applied to the AIRS/AMSU FOV. We define a *cloud scene* as a list of the cloud types that are present within a given AIRS/AMSU FOR. For example, the notation (Ci,Ac,Sc,Cu) is used to label a cloud scene that contains cirrus, altocumulus, stratocumulus, and cumulus clouds at cloud top in any frequency and in any sequence along the orbit segment. These simplifications greatly reduce the dimensionality of the classification problem and makes cloud scene identification tractable. We will show both partly cloudy and completely cloudy scenes in Section 4, so the clear sky (nc) type is both included and excluded in the analyses. Since each of the 8 cloud types is either present or absent, a cloud scene can also be represented by an 8 bit binary string. As a consequence, there are 256 ($2^8$) possible cloud scenes that remain after taking into account the aforementioned simplifications. The number of possible cloud scenes is therefore reduced from $9^{50 \times 125}$ to a much more tractable 256. The limitations of this approach are: a consideration of cloud tops only, the spatial sequence and frequency of individual cloud types are not considered, and equal weight is given to all cloud types within a cloud scene regardless of counts.

One advantage of using a classification to define cloud mixtures rather than an unsupervised learning technique, such as clustering, is that the size of the set of possible cloud mixtures is well defined and finite (here it is 256). A related and important advantage of classification is that one can use this set of classes (cloud scenes) with any parameter matched to any given scene. Here, the spatial scale dependence of those cloud scenes is described in section 3.2.

An alternative approach may consider the vertical layering of cloud types or cloud features, some form of weighting based on counts of each cloud type, or possibly the sequence of cloud types, which may result in different radiance measurements observed by the AIRS instrument (the radiance emitted within an AIRS footprint is non-uniform and channel-dependent, as described in Schreier et al., 2010). However, the simplified approach outlined above is broadly consistent with the sensitivity and sampling characteristics of nadir-viewing passive infrared sounders, yet simplifies and makes tractable the complexity of cloud scenes. Therefore, we consider the approach outlined above to be an appropriate compromise that retains the diversity of cloud scenes and makes the necessary data processing tractable by reducing the dimensionality for ease of interpretation.

## 2.3 AIRS thermodynamic phase and ice cloud properties

In this study, the AIRS version 6 cloud thermodynamic phase and cloud ice properties (Kahn et al., 2014) are geolocated to the CloudSat ground track and are binned by cloud scene. The cloud thermodynamic phase algorithm includes two liquid tests and four ice tests of brightness temperature ($T_b$) thresholds and $T_b$ differences ($\Delta T_b$) in the mid-infrared atmospheric windows. The $T_b$ and $\Delta T_b$ thresholds are designed to exploit spectral differences in liquid and ice water indices of refraction. The two



liquid and four ice tests are each assigned a value of -1 and +1, respectively, and a summed value that ranges from -2 to +4 is reported. Summed values −2 or −1 indicate liquid clouds, 0 is undetermined, and ≥ +1 indicate ice, with the highest values indicating deeper, convective ice clouds (Naud and Kahn, 2015). Ice is detected in 26.5% of AIRS footprints by Kahn et al. (2014) and pixel-scale comparisons with estimates of ice from the Cloud-Aerosol Lidar and Infrared Pathfinder Satellite

5  Observation (CALIPSO) lidar (Jin and Nasiri, 2014) are in agreement with AIRS more than 90% of the time. The success rate, however, is smaller for liquid cloud detection with AIRS using CALIPSO as a benchmark because of the small thermal contrast between low-lying liquid clouds and the surface. Despite this limitation in sensitivity, AIRS rarely misidentifies liquid clouds as ice (Jin and Nasiri, 2014). Furthermore, many liquid clouds are classified as undetermined phase. Low latitude shallow trade cumulus clouds generally fall within this category (Kahn et al., 2017).

Kahn et al. (2014) describe a retrieval algorithm that is based on optimal estimation (OE) theory and derives ice cloud optical thickness ($\tau_i$) and effective radius ($r_{ei}$) for AIRS footprints containing ice. Scalar averaging kernels (AKs), $\chi^2$ residuals from observed and simulated radiance fits, and values of relative error are also reported (Kahn et al., 2014). Values of AKs closer to 1.0 suggest higher information content while larger relative error estimates and values of $\chi^2$ indicate increased uncertainty

in retrieved parameters. Only the relative magnitude of error estimates should be considered since temperature, specific humidity, surface temperature, surface emissivity, and ice crystal habit and size distribution uncertainties are not included in the error covariance matrices of the AIRS version 6 algorithm (cf. Kahn et al., 2014). We focus on the differences in error estimates and $\chi^2$ among cloud scenes and determine which cloud scenes contain higher or lower certainty in their ice cloud properties relative to other scenes.

**3 Classification and characteristics of cloud scenes**

**3.1 Cloud scenes with ~45 km resolution**

A cloud scene is assigned to every AIRS/AMSU FOR along the CloudSat viewing path using the methodology outlined in Section 2. Using the two years of data, a total of 194 out of 256 possible cloud scenes are observed but only 18 of the cloud scenes account for 90% of all observed scenes (Fig. 1). The four most common scenes contain one cloud type with or without

clear sky, and the most common mixed cloud scene (Ac, Sc) is ranked as the fifth most common scene overall. Intuitively, the more diverse a scene, the less frequently it should be observed. The scene that ranked last (18[th]) within the 90[th] percentile is (Ci, Ac, Sc). The least frequently observed cloud scene with a ranking of 194  contains six cloud types (Ci, As, Ac, St, Cu, Ns) and was observed only once in two years. Of the 256 possible types of cloud scenes, the number of unobserved cloud scenes is 62, of which 61 include St. The unobserved cloud scenes include the only possible cloud scene with eight cloud types

together, and the seven possible cloud scenes with seven cloud types together.



The unobserved scenes in the two-year period contain a median of five different cloud types. This is consistent with the improbability of particular cloud types occurring in rapid succession over a few tens of kilometers. The only unobserved cloud scene that does not contain St is (Sc,Cu,Ns,Dc) and is consistent with the conclusion by Sassen and Wang (2008) that Dc (1.8%) and Cu (1.7%) clouds are the least frequent of the cloud types. While Dc and Ns are typically associated with different
climatological regimes (tropical convection versus extratropical storm tracks), occasionally, Dc is embedded within extratropical cyclones and Ns is classified in stratiform regions of mesoscale convective systems (MCSs). Given the prevalence of Sc and Cu in Figure 1, it is somewhat surprising that the combination (Sc,Cu,Ns,Dc) is not observed.

Figure 2 depicts the geographic distribution of Sc and clear, the two most observed scenes, and (Ac,Sc) the most observed
mixed cloud scene. The pattern of clear sky scenes is generally similar to the clear sky global distribution derived by Eguchi and Yokota (2008) from CALIPSO and Moderate Resolution Imaging Spectroradiometer (MODIS) data. The Sc classification is consistent with the prevalence of low clouds in subsidence regions and trade cumulus in the tropics and subtropics (e.g., Yue et al., 2011). The (Ac,Sc) cloud scene is identified most frequently in the extratropical storm tracks and the transition from subtropical cumulus to deep tropical convection.

**3.2 Cloud scenes at 1 to 1000 km scales**

To investigate the scale dependence of the number of cloud scenes, the approach described in Section 2 is modified for a range of horizontal extents between 1.1 and 1000 km. The number of observed cloud scenes calculated at each horizontal scale is shown in Fig. 3a for 10 to 1000 km. At the finest scale of 1.1 km, only eight possible observed cloud scenes or clear sky are expected. When the scale increases, as expected, the number of cloud scenes quickly increases with a total of 143 cloud scenes
observed at a scale of 11 km. As horizontal scale is further increased, the probability of observing cloud scenes with only one or two cloud types is reduced. After a maximum number of cloud scenes is obtained at 105 km, the number of cloud scenes will decrease with increasing scale (e.g., 163 cloud scenes at 990 km) until a limiting case is reached at the largest scale with only one cloud scene with all observed cloud types. The number of cloud scenes observed at least once at the AIRS/AMSU FOR horizontal scale (indicated by the red vertical line on Fig. 3a) is approximately 190.

The 90[th] percentile calculated at all horizontal scales is shown in Fig. 3b. The 90[th] percentile of the maximum number of cloud scenes is 33 at 303 to 440 km horizontal scale. The number at the nominal 45-km AMSU footprint scale is 16 cloud scenes, while the average number at the AIRS footprint is 9 cloud scenes. While these results show that fewer cloud type mixtures are observed at decreasing lengths of 45 km to 15 km, a large variety of cloud type mixtures is still encountered. While infrared
sounding at 15 km resolution does not eliminate the cloud scene complexity encountered for combined infrared and microwave sounding at 45 km, the vast majority of 15 km footprints contain a smaller subset of possible cloud mixtures. In Section 4, we will determine whether individual cloud types or cloud type mixtures have meaningful impacts on AIRS cloud property retrievals. (Impacts on temperature and specific humidity soundings are beyond the scope of this initial study.)




The maximum number of observed cloud scenes (210) at a particular horizontal scale (105 km) remains unexplained. The scale preference depends on the physical characteristics of cloud regimes and the degree to which cloud types are "mixed together" by region and furthermore depend on cloud length distributions (Guillaume et al., 2018). A simple model is described

below that is able to approximate the results of Fig. 3.

### 3.3 Generalizing to all scales: why a maximum number of cloud scenes at 105 km?

The goal of this section is to derive cloud scene scale statistics that are independent of any regular grid resolution, and explore whether these statistics can explain some features of the number of scenes as a function of scale observed in the previous section. In particular, we explore whether these statistics can explain the maximum observed around 105 km. There is however

an inherent difficulty in defining the boundaries that delimit any given cloud scene in absence of a predefined horizontal extent. It is possible that within a given cloud scene there exists several scenes with the same cloud types but differing lengths making the scene identification ambiguous. To circumvent this problem, we define a cloud scene and its maximum length as follows:

    (1)  We search for a cloud scene containing a pre-defined mixture of cloud types. The spatial extent of this scene is

delimited by cloud types (or clear sky) on both ends that do not belong to the mixture.

    (2)  The maximum length of a cloud scene is the sum of all the horizontal lengths of all the cloud types in the cloud scene.

For example, imagine that we will calculate the maximum length of the specific cloud scene (Ac,Sc). We then identify a

location in the CloudSat data record with the following illustrative succession of cloud types: (Ci,Ac,Sc,Ac,Sc,Ac,Ns), with the number of CloudSat profiles associated with each cloud type of 10, 3, 6, 5, 7, 12, and 15, respectively. The Ci and Ns obviously do not belong to the (Ac,Sc) cloud scene and therefore delimit the scene as defined in (1) above. The maximum length of the cloud scene (Ac,Sc) will be the sum of the number of CloudSat profiles for (Ac,Sc,Ac,Sc,Ac), which is 3+6+5+7+12=33 CloudSat profiles in total. Below, we define a minimum cloud length that is unequivocal:

    (3)  If within a given cloud scene, there exist several cloud scenes with the same cloud types but smaller lengths than the maximum length, the minimum length of a cloud scene is defined as the smallest length of all those lengths.

In the example above, there are four possible sequences (in bold font) that could be the minimum length: (**Ac,Sc**,Ac,Sc,Ac),

(Ac,**Sc,Ac**,Sc,Ac), (Ac,Sc,**Ac,Sc**,Ac) or (Ac,Sc,Ac,**Sc,Ac**). The corresponding lengths are 3+6=9, 6+5=11, 5+7=12, and 7+12=19, respectively. In this example, the minimum length would therefore be 9 CloudSat profiles. (The minimum and maximum may be equal for a particular mixed cloud scene.)



Before steps (1) and (2) are used to quantify the maximum and minimum lengths for each of the 247 mixed scenes (256 minus the 8 single cloud scenes and clear sky), the locations of each cloud scene must first be identified in the two-year data record. Starting at the first CloudSat profile, the presence of each of the 247 mixed cloud scenes is determined using (1). For each occurrence of each mixed cloud scene, (2) and (3) are then applied to determine the maximum and minimum lengths for each
individual cloud scene. After processing the maximum and minimum lengths for every mixed cloud scene, simple statistics are calculated.

A total of 200 out of 247 possible mixed scenes were identified. The minimum and maximum length occurrence frequencies of five cloud scenes (AlCu,Sc), (AlSt,Sc,Cu), (ci,AlSt,Cu,DC), (AlSt,AlCu,Ns,DC) and (ci,AlSt,AlCu,St,Sc) selected
randomly from the 200 present in the two-year record, are shown in Fig. 4a and 4c, respectively. From top to bottom, their respective ranks are 1, 26, 51, 76 and 101. It is striking that each frequency histogram in Fig. 4a and 4c is not monotonic and displays a frequency maximum between 100 and 1000 km. Consequently, the sum of all (200) observed mixed scenes across length scales will result in a curve with a maximum and these are shown in Fig. 4b and 4d. Both curves are very similar to Fig. 3 and have maxima for about 180 observed scenes at 77 km and 174 km, respectively. This demonstrates that the scale
dependence of the number of observed scenes is the result of the scale dependence of frequency distributions of each mixed cloud scene, and that the maximum of the number of observed scenes will occur between 77 and 174 km, depending on the length of the cloud scene.

In order to shed additional light on why a maximum in the occurrence frequency of each cloud scene histogram is obtained, histograms of cloud length frequency of *single* cloud types (defined at cloud top) are calculated. An example CloudSat orbital
segment is shown in Fig. 5. The distribution of lengths for each cloud type for the two-year period is then shown in Fig. 6 with corresponding mean and median values reported in Table 1. Note that these values are similar to but not exactly the same as those calculated in Guillaume et al. (2018), for which cloud length was derived from a 2-D curtain of cloud features. The main characteristic shared by all cloud types in Fig. 6 is that their distributions are heavily skewed towards small lengths.

The length of a mixed scene is the sum of the lengths of each cloud type within it. There are two aspects that will influence
the number of scenes observed at a given length L. First, there are several combinations of different lengths that will sum to L and those lengths will be smaller than L (abscissa of Fig. 6). Second, the likelihood of observing a given scene depends on the frequency of occurrence of each cloud type (ordinate axis of Fig. 6). These two effects have opposite behaviors as a function of L: single cloud frequency decreases with L, whereas the number of cloud length combinations that sum up to L increases with length scale.
To illustrate the effects of these opposing behaviors, we consider the scene (AlSt,Sc,Cu) length distribution. Since the minimum length of all cloud distributions in Fig. 6 is one CloudSat profile, there is only one possible cloud length combination (1+1+1) that will sum to the minimum possible length of the scene (AlSt,Sc,Cu). This is indeed the value observed on the far left of each red-orange curve in Fig. 4a and 4c. Next, consider a measurement consisting of 4 CloudSat profiles with this particular scene, with three possible length combinations: (1+1+2), (1+2+1) or (2+1+1). The frequency of each individual





cloud type, AlSt, Sc or Cu, is smaller at the scale of 4 CloudSat profiles than it is at a length of 3 CloudSat profiles in Fig. 6. However, there are more (AlSt,Sc,Cu) scenes at length 4 than at 3 in Fig. 4a and 4c. This indicates that the increase of possible combinations is more important than the individual cloud frequency decrease for larger scales. This reasoning applies for increasing lengths until the decreasing frequency of individual cloud types between two consecutive lengths is more important.

There are very few single AlSt, Sc or Cu clouds observed at large lengths (far right scale of Fig. 6) resulting in a very small number of observed (AlSt,Sc,Cu) scenes in Fig. 4a and 4c, despite the large number of length combination possibilities that may contribute.

To summarize, we showed that the scale dependence of the number of mixed scenes is the result of two opposing contributions. First, the single cloud type cloud lengths distributions are heavily skewed toward small scales (Fig. 6), qualitatively explaining

the scale dependence of a mixed scenes cloud lengths distributions (Fig 4a and 4c). Secondly, the scale dependence of all observed mixed scenes cloud lengths distributions (Fig 4a and 4c) explains the scale dependence of the number of mixed scenes (Fig 4b and 4d).

## 4 Cloud scene dependence of AIRS cloud properties

We will now establish differences in the AIRS thermodynamic phase and ice cloud properties in the presence of complex and simple cloud types using coincident cloud scenes. The AIRS cloud thermodynamic phase tests are discussed separately for single and mixed cloud scenes in Section 4.1. The AIRS ice cloud $\tau_i$ and $r_{ei}$, error estimates, averaging kernels (i.e., information content), and $\chi^2$ residual fits between observed and simulated radiances are shown in Section 4.2.

### 4.1 Cloud thermodynamic phase

The occurrence frequencies of the sum of all phase tests ranges between -2 and +4 for cloud scenes that contain only single cloud types (no clear sky and no mixtures of cloud types) are shown in Fig. 7. (While these clouds account for a small percentage of the total number of AIRS FOVs, homogeneous cloud scenes serve as an ideal point of reference for establishing cloud phase sensitivity benchmarks.) The ice tests dominate the Ci cloud scenes and reaffirm the sensitivity of AIRS to ice clouds. CloudSat-classified clear scenes contain occasional occurrences of AIRS-detected thin cirrus (+1 and +2), consistent

with either thin cirrus that is undetected by the CloudSat radar, or thicker cirrus within the AIRS footprint but to the side of the CloudSat ground track (e.g., Kahn et al., 2008). A few occurrences of -1 and -2 may also arise from spatial mismatches between AIRS and CloudSat scenes, or from stratus below 1 km in altitude that is undetected by CloudSat. In the Sc cloud scenes, trade cumulus clouds dominate as previously shown by Yue et al. (2011) and Kahn et al. (2017). A larger proportion of liquid tests, and a smaller proportion of ice tests, is observed in the Sc cloud scenes compared to clear sky, but undetermined

phase is dominant in both scene types. The Cu and Sc cloud scene histograms are generally similar with more undetermined cases for Cu, but with a slight reduction of liquid and slight increase in ice observed for Cu compared to Sc.



The As cloud scene histogram in Fig. 7 is overwhelmingly dominated by ice. The undetermined cases in part may result from supercooled liquid or mixed-phase clouds that potentially could be distinguished with an improved phase algorithm that factors in the spectral mid-infrared signature of supercooled liquid (e.g., Rowe et al., 2013). The Ac and As cloud scene histograms

are very different from each other, with a majority of undetermined and liquid for Ac and a majority of ice for As, consistent with aircraft observations (Mazin, 2006). The preponderance of undetermined phase for Ac may indicate frequent supercooled liquid cloud tops (Zhang et al., 2010). Ham et al. (2013) showed that Ac are typically 2-3 km lower in altitude than As and this probably explains some of the difference in liquid and ice phase, as lower clouds are usually warmer. The Ns cloud scene histogram is dominated by ice detection with occasional liquid and undetermined cloud tops. The Ns cloud scene also has

significant height overlap with Ac and As, with most tops for all three types typically located below 9 km. Ice tests dominate in the Dc cloud scene histogram although a very small proportion of -1, 0, and +1 occur. Inspection of AIRS granules (not shown) demonstrate that the spectral signatures used in thermal infrared phase tests break down in the presence of overshooting convection and other ice clouds within a few Kelvins of the tropopause (e.g., Kahn et al., 2018).

The nine most frequent cloud scenes in Fig. 1 are shown in Fig. 8. While a few of these cloud scenes are marked as single cloud types, we include clear sky here, unlike those shown in Fig. 7. In the case of Cu, Ns, As, and Sc cloud scenes, the histograms are very similar when comparing Figs. 7 and 8. In the Ci cloud scene histograms, there is a small but notable reduction in the frequency of ice tests and an increase in liquid and undetermined in Fig. 8 compared to Fig. 7. This is consistent with the expectation that more heterogeneous Ci cloud scenes with mixtures of clear sky exert a weaker spectral infrared phase

signature. In the Ac cloud scene histograms, there is a small but discernible increase in ice tests in Fig. 8 compared to Fig. 7. Horizontally heterogeneous Ac therefore has a more robust ice signature than horizontally homogeneous Ac. The (Ci,Sc) cloud scene ice phase histogram resembles a hybrid of histograms for Ci and Sc with undetermined phase the most frequent. (Ci,Sc) is a common cloud scene in the low latitudes as trade cumulus (Sc cloud type) and is frequently found under thin cirrus (Chang and Li, 2005). Furthermore, the spectral signatures of the two types of clouds frequently cancel, giving an undetermined phase

result in the spectral tests used here (not shown). The (Ci,As) cloud scene shows a slight reduction in putative liquid detections and a slight increase in ice detections compared to As alone. While the As cloud scene in Fig. 7 is dominated by +2, the (Ci,As) cloud scene is dominated by +2 and +3. This suggests that a mixture of Ci and As together can trigger more ice tests in AIRS than As alone. Lastly, the mixed (As,Sc,nc) cloud scene in Fig. 8 closely resembles pure Ac and Sc cloud scenes in Fig. 7 since all three histograms are similar.

There is strong differentiation in the cloud thermodynamic phase among cloud scenes with single cloud types. Ice tests dominate Ci, Ns, Dc, and As, while liquid and undetermined tests dominate Ac, Sc, and Cu. In most mixed cloud scenes, the characteristics of the histograms are similar either to single types or have combined characteristics of the multiple cloud types contained within the cloud scene. These differences between single and mixed cloud scenes can be explained by the fact that



the AIRS footprint ( ~ 15 km ) is commensurate with the dimension of a single cloud so that the most frequent observations involve the characteristics of one cloud. Cloud lengths statistics, such as median lengths and median absolute deviations calculated from the data shown in Fig. 6, corroborate this statement. These results are encouraging and reaffirm the capabilities of thermal infrared cloud phase determination (Jin and Nasiri 2014) and exhibit consistency with cloud types from the CloudSat

radar. Furthermore, the AIRS cloud phase determination is not adversely affected by complex cloud top morphology (but without considering vertical structure). We note, however, that the AIRS phase determination has some ambiguity in overlapping ice and liquid cloud layers as previously shown by Jin and Nasiri (2014).

**4.2 Ice cloud properties**

The mean ice cloud property retrievals are summarized in Table 2 for cloud scenes composed of single cloud type only and

are depicted in Fig. 7. The Sc cloud scene shows a small population of cirrus that go undetected in 2B-CLDCLASS (Fig. 7) and have mean values of $\tau_i$ =1.36 and $r_{ei}$=20.3 µm (Table 2).  The AKs are also lowest in Table 2 for Sc relative to other cloud scenes with similarly high errors in $\tau_i$ and $r_{ei}$. Kahn et al. (2008; 2015) have shown that AIRS is very sensitive to thin cirrus, thus some ice clouds in CloudSat-identified Sc cloud scenes are expected. Because tenuous ice clouds have smaller values of $\tau_i$ and $r_{ei}$, the lower estimates of information content and larger error estimates are promising. These tenuous ice cloud retrievals

are differentiated well from more robust retrievals within cloud scenes that are dominated by ice phase in the histograms (Fig. 7 and Table 2).

The Ci cloud scene has mean values of $\tau_i$ =1.94 and $r_{ei}$=25.7 µm, an AK=0.99, the highest of any scene type, and lower errors compared to other types in Table 2. The As cloud scene has a larger mean of $\tau_i$=2.55, a slightly smaller error, and slightly

larger AK than the Ac cloud scene (Table 2). Interestingly, the mean value and error estimate of $r_{ei}$ is lower for Ac than As, exhibiting differentiation between these two mid-level cloud types. However, a much smaller proportion of Ac are ice compared to As (Fig. 7).

The Ns cloud scene in Table 2 contains larger mean values of $\tau_i$ and $r_{ei}$ than Ci and Ac cloud scenes, but these values are similar

to those for As; however, the mean values are lower than Cu and Dc cloud scenes (Table 2). A lower value of $\tau_i$ is characteristic of diffuse cloud tops where the infrared emission may originate several km deep within the cloud (e.g., see Kahn et al., 2008; Holz et al., 2008). The reduced AK=0.89 for the Ns cloud scenes shows that a diffuse cloud top is more problematic for ice cloud retrievals.  Cu cloud scenes with ice cloud tops occur a small amount of the time (Fig. 7); furthermore, Cu is infrequent in CloudSat classification (1.7% of all clouds). The horizontal extent of Cu is also much smaller than Dc (see Table 1).

Interestingly, $\tau_i$ is larger for Cu cloud scenes than for all categories except Dc cloud scenes (Table 2).

The mean Cu value of $r_{ei}$=27.9 µm is the largest for all cloud scenes. This is consistent with larger ice particles observed at the tops of convection instead of small ice particles in thin cirrus at the same cloud top temperature (e.g., Yuan and Li 2010; Protat





et al. 2011; van Diedenhoven et al. 2014; Kahn et al., 2018). These Cu cases are likely transient cumulus congestus at altitudes cold enough for cloud top glaciation. The Dc cloud scene has the largest mean $\tau_i$=5.54 of all cloud scenes with a very dense cloud top that saturates the infrared emission signal in contrast to Ns. The values of $r_{ei}$ for Dc are nearly as large as Cu (Table 2) with a slight reduction in the $r_{ei}$ AK=0.96 and $\tau_i$ AK=0.98 relative to Ci cloud scenes (Table 2). This is consistent with
reduced sensitivity for high values of $\tau_i$ (e.g., Huang et al., 2004).

The relative variations between the ice cloud retrieval properties for single type cloud scenes in Table 2 are consistent with expectations of infrared sensitivity. CloudSat-observed Ci cloud scenes have smaller error estimates and higher information content in comparison to Sc, consistent with Sc containing tenuous cirrus that goes undetected by CloudSat. Larger $\tau_i$ and $r_{ei}$
are observed at the tops of convective ice clouds such as Dc and Cu compared to stratiform clouds such as As and Ci. Differences in ice cloud properties between Ac and As cloud scenes are consistent with observed differences in scene heterogeneity and cloud top height.

The mean ice cloud property retrievals for clear sky, mixed cloud scenes with or without clear sky, and cloud scenes with one
cloud type mixed with clear sky, are summarized in Table 3. Scenes identified as clear sky exhibit properties of a small population of thin cirrus detected by AIRS (Fig. 7) with mean values of $\tau_i$ =0.78 and $r_{ei}$=20.9 μm (Table 3). The AKs are notably lower than other cloud scenes and the relative error for $\tau_i$ is higher.The corresponding cloud phase histograms are shown in Fig. 8. The (Sc,Nc) cloud scene in Table 3 is similar to the Sc cloud scene in Table 2 except the (Sc,Nc) $\tau_i$ is slightly reduced in Table 3. This is consistent with some thin cirrus in the (Sc,Nc) cloud scene, expected since thin cirrus and shallow
Cu frequently overlap in the subtropical and tropical trade cumulus regimes. The (As,Nc) and (Ac,Nc) cloud scenes in Table 3 are very similar to As and Ac cloud scenes in Table 2 except for slight reductions in $\tau_i$ and AK for $r_{ei}$ and $\tau_i$, and a slight increase in their estimated errors. Scenes in Table 3 are partly cloudy, implying a weaker infrared cloud signal. The slight degradation in the fidelity of ice cloud for retrievals (As,Nc) and (Ac,Nc) is confirmed by small reductions in AKs and increases in $\chi^2$ compared to the fully cloud scenes in Table 2. The (Ci,Nc) cloud scene in Table 3 shows only slight changes
from the Ci cloud scene in Table 2, consistent with some scenes containing thin cirrus undetectable by CloudSat and flagged as clear.

The differences between Tables 2 and 3 are more significant for the convective ice clouds, however. The (Cu,Nc) cloud scene $\tau_i$ and AK are smaller while errors are larger in Table 3 compared to the pure Cu cloud scene in Table 2. This is expected as
Cu clouds are several kilometers in depth but often have small horizontal scales and are averaged with clear sky in an AIRS pixel. Mixed (Cu,Nc) cloud scenes are especially problematic for plane-parallel radiative transfer calculations. This results in more uncertain retrievals of ice cloud properties for partial cloud (Cu,Nc) in Table 3 than those for the pure Cu cases in Table 2, which are more likely to be completely fill an AIRS scene. Recall that any combination of Cu and Nc qualifies as (Cu,Nc), including mostly clear ones.





To summarize Tables 2 and 3, larger differences in ice cloud property retrievals are found between different cloud types than between pure and mixed cloud scenes. The AIRS cloud property retrievals are not greatly affected by mixtures of cloud types within the AIRS footprint, and ice cloud property differences among cloud scenes are broadly consistent with the expected

performance of infrared retrievals among cloud types.

## 5 Summary

A method is described to classify cloud mixtures of cloud top types, termed *cloud scenes,* using the 2B-CLDCLASS cloud type classification obtained from the 94 GHz CloudSat radar. The scale dependence of those cloud scenes is studied. The method is initially applied to two years of CloudSat data collocated within the Atmospheric Infrared Sounder

(AIRS)/Atmospheric Microwave Sounding Unit (AMSU) field of regard (FOR) at 45 km scale. Given that 45 km scale and approximately 50 coinciding CloudSat profiles, each with 125 levels, the total number of possible scenes within an AIRS/AMSU FOR is $9^{50 \times 125}$. This very large number of possible scenes is reduced to 256 by making three assumptions in the classification. First, only the cloud type at the cloud top is considered. Second, the occurrence frequency of each cloud type within the cloud scene is disregarded; thus, there is no consideration of the counts of each cloud type. Third, the sequence of

cloud types along the orbit segment is not considered. These three assumptions make mixed cloud scene classification tractable and are broadly consistent with the sensitivity of infrared sounders to clouds. They are also independent of the spatial scale of a scene and therefore can be generalized to all horizontal scales. A total of 194 out of 256 possible cloud scenes are observed in a two-year period from 01 July 2006 to 30 June 2008. The maximum number of cloud scenes occurs at a horizontal scale of 105 km with fewer cloud scenes at larger and smaller scales, and the majority of observed cloud scenes contain single cloud

types.

Summarizing AIRS cloud top property retrievals for scenes with no clear sky (Table 2 and Figure 7), there is strong differentiation in the cloud thermodynamic phase for scenes with only one cloud type. Ice phase dominates Ci, Ns, Dc, and As, while liquid and undetermined phase dominate Ac, Sc, and Cu. While not a direct comparison, in most mixed cloud scenes

the changes in the ice properties are generally either small or reflect the combined characteristics of the multiple cloud types contained within the cloud scene. The sensitivity of thermal infrared cloud phase determination is consistent with independently determined cloud typing from the CloudSat radar for clouds detected by CloudSat. Furthermore, the cloud phase determination from AIRS is not adversely affected by more complex cloud scene morphology as indicated by CloudSat cloud type determined at the cloud tops. The relative magnitude of differences in $r_{ei}$ and $\tau_i$, and their averaging kernels (AKs) and

error estimates, and the $\chi^2$ residual between simulated and observed radiances, are consistent with expectations of infrared retrieval sensitivity to different cloud types. Smaller error estimates and higher information content (AKs) within Ci cloud scenes are observed in comparison to thin cirrus likely missed by CloudSat in clear sky and Sc scenes. Larger $\tau_i$ and $r_{ei}$ are



observed at the tops of convective ice clouds. Differences in retrieved cloud properties between Ac and As cloud scenes are consistent with differences in their scene heterogeneity and cloud temperature. Variations in ice cloud property retrievals are larger between types of cloud scenes than between cloudy and partly cloudy/mixed cloud scenes.

5   The fidelity of AIRS retrieved cloud phase and ice cloud microphysics was tested within uniform and mixed scenes. As with phase, retrieval differences are shown to be larger among cloud types rather than between uniform and mixed cloud scenes. While not an emphasis of this particular investigation, previous studies have shown that phase identification (Jin and Nasiri 2014), ice microphysical retrievals (e.g., Kahn et al., 2015), and thermodynamic profile soundings – especially within the planetary boundary layer (e.g., Irion et al. 2018) – will remain a significant challenge for multi-layer clouds.

New methodologies for simultaneous retrievals of cloud microphysical properties and temperature and specific humidity profiles that include clouds in the forward radiative transfer (e.g., De Souza-Machado et al. 2018; Irion et al. 2018) necessitate careful investigation of the effects of cloud mixtures on retrieved cloud properties. The bias and root-mean square error of AIRS temperature and specific humidity soundings depend on cloud type (Yue et al., 2013; Wong et al., 2015). A more rigorous

evaluation of scene complexity is necessary for optimizing the retrieval configuration of future sounding algorithms (Irion et al. 2018) and for validating their products.

This investigation shows that careful inspection of footprint-scale AIRS cloud property retrievals are consistent with expectations of infrared sensitivity to different cloud types defined with the 94 GHz CloudSat radar. Other cloud observations, such as MODIS, may be used in a similar analysis to the one described here. MODIS captures the the off-nadir portion of the

AIRS swath and the fine-scale variability within AIRS footprints. Wang et al. (2016) used the cloud typing in CloudSat to cross validate with cloud typing using MODIS-defined cloud types. This establishes a link between cloud types obtained from CloudSat and MODIS. A rigorous estimation of the pixel-scale relationships between cloud properties obtained from CloudSat, MODIS and AIRS/AMSU will help to further advance multi-sensor and multi-variate geophysical retrievals (e.g., Irion et al.

2018).

**Acknowledgments**

Part of this research was carried out at the Jet Propulsion Laboratory (JPL), California Institute of Technology, under a contract with the National Aeronautics and Space Administration. This project was supported by NASA's Making Earth Science Data Records for Use in Research Environments (MEaSUREs) program. CloudSat data were obtained through the CloudSat Data

Processing Center (http://www.cloudsat.cira.colostate.edu/). The combined data files used in this work are available at the Goddard Earth Sciences Data and Information Services Center (ftp://measures.gsfc.nasa.gov/data/s4pa/AIRS_CloudSat/). ©️ 2018. All rights reserved. Government sponsorship acknowledged.



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



| cloud_type | nc | Ci | AlSt | AlCu | St | Sc | Cu | Ns | DC |
|---|---|---|---|---|---|---|---|---|---|
| median | 6.6 | 8.8 | 12.1 | 1.1 | 3.3 | 1.1 | 1.1 | 15.4 | 14.3 |
| mad | 25.7 | 29.6 | 39.2 | 11.1 | 6.1 | 9.8 | 6.0 | 42.8 | 28.9 |

**Table 1. Horizontal cloud chord length median and median absolute deviation (m.a.d.) for each cloud type, in kilometers.**





| Cloud type | Single cloud type proportion | Mean $\tau_i$ | $\tau_i$ relative error | $\tau_i$ averaging kernel | % passing QC for $\tau_i$ | Mean $r_{ei}$ | $r_{ei}$ relative error | $r_{ei}$ averaging kernel | % passing QC for $r_{ei}$ | $\chi^2$ residual fit |
|---|---|---|---|---|---|---|---|---|---|---|
| Ci | 25.2 | 1.94 | 1.99 | 0.99 | 96.5 | 25.7 | 2.6 | 0.99 | 73.6 | 4.1 |
| As | 26.6 | 2.55 | 5.55 | 0.94 | 97.7 | 25.0 | 4.6 | 0.98 | 80.2 | 2.9 |
| Ac | 5.5 | 1.60 | 5.94 | 0.92 | 94.3 | 22.2 | 3.6 | 0.99 | 57.0 | 4.2 |
| Sc | 22.2 | 1.36 | 14.17 | 0.72 | 78.5 | 20.3 | 6.7 | 0.96 | 48.4 | 3.4 |
| Cu | 1.0 | 3.27 | 5.29 | 0.96 | 93.5 | 27.9 | 7.1 | 0.96 | 68.8 | 3.5 |
| Ns | 15.4 | 2.52 | 8.21 | 0.89 | 97.8 | 23.9 | 5.6 | 0.98 | 86.7 | 2.5 |
| Dc | 4.0 | 5.54 | 3.54 | 0.98 | 98.4 | 27.2 | 7.2 | 0.96 | 72.5 | 3.1 |

**Table 2. Cloud ice properties for single cloud types (all CloudSat pixels within one AIRS footprint have the same cloud type). Proportions and relative errors are in percent. The effective radius is in μm.**





| Cloud scene | Mixed scenes proportion | Mean $\tau_i$ | $\tau_i$ relative error | $\tau_i$ averaging kernel | % passing QC for $\tau_i$ | Mean $r_{ei}$ | $r_{ei}$ relative error | $r_{ei}$ averaging kernel | % passing QC for $r_{ei}$ | $\chi^2$ residual fit |
|---|---|---|---|---|---|---|---|---|---|---|
| nc | 30.7 | 0.78 | 14.8 | 0.71 | 75.2 | 20.9 | 5.6 | 0.97 | 34.3 | 3.3 |
| Sc | 24.7 | 0.98 | 14.5 | 0.71 | 75.0 | 21.1 | 5.4 | 0.97 | 38.2 | 3.4 |
| As | 9.8 | 2.35 | 6.8 | 0.91 | 97.0 | 24.3 | 4.8 | 0.98 | 78.2 | 2.9 |
| Ci | 9.5 | 1.71 | 2.5 | 0.98 | 94.9 | 25.0 | 2.7 | 1.0 | 71.9 | 4.0 |
| Ns | 5.3 | 2.41 | 8.7 | 0.89 | 98.1 | 23.6 | 5.6 | 0.98 | 87.0 | 2.5 |
| Ac | 3.0 | 1.30 | 6.3 | 0.91 | 90.9 | 21.5 | 3.8 | 0.98 | 55.7 | 4.3 |
| Ac, Sc | 2.6 | 1.00 | 7.6 | 0.89 | 85.3 | 21.5 | 4.3 | 0.98 | 54.6 | 4.3 |
| Ci, As | 1.8 | 2.46 | 4.2 | 0.96 | 97.3 | 24.6 | 3.8 | 0.99 | 73.4 | 3.5 |
| Cu | 1.5 | 1.60 | 10.4 | 0.82 | 80.5 | 23.5 | 5.6 | 0.97 | 48.1 | 3.8 |
| Ci, Sc | 1.3 | 0.79 | 4.2 | 0.95 | 85.9 | 23.2 | 3.0 | 0.99 | 61.0 | 4.3 |

**Table 3. Cloud ice properties for the mixed cloud scenes in the 90[th] percentile of all observed cloud scenes. Relative errors are reported in percentage. Proportions and relative errors are in percent. The effective radius is in μm.**





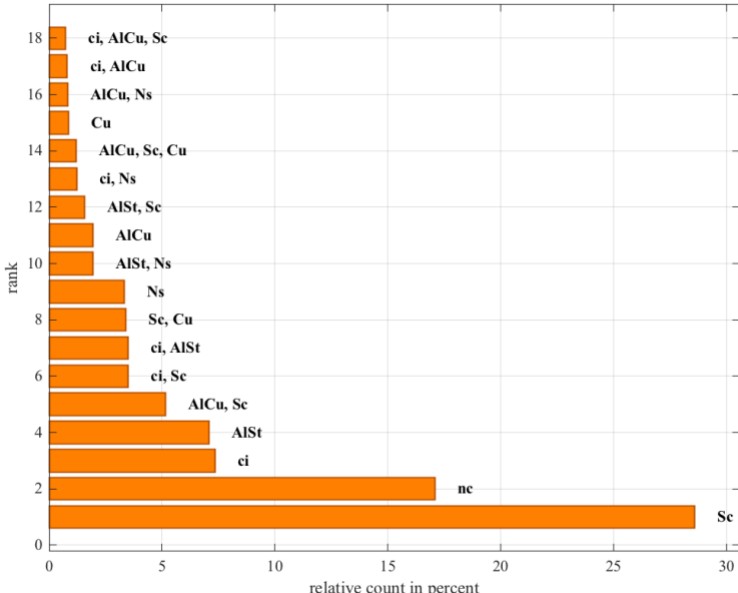

**Figure 1.** Histogram of cloud scenes relative counts of occurrence. The cumulative sum of the relative counts of these 18 cloud scenes
amounts to (at least) 90% of all cloud scenes observed globally over a period of two years.





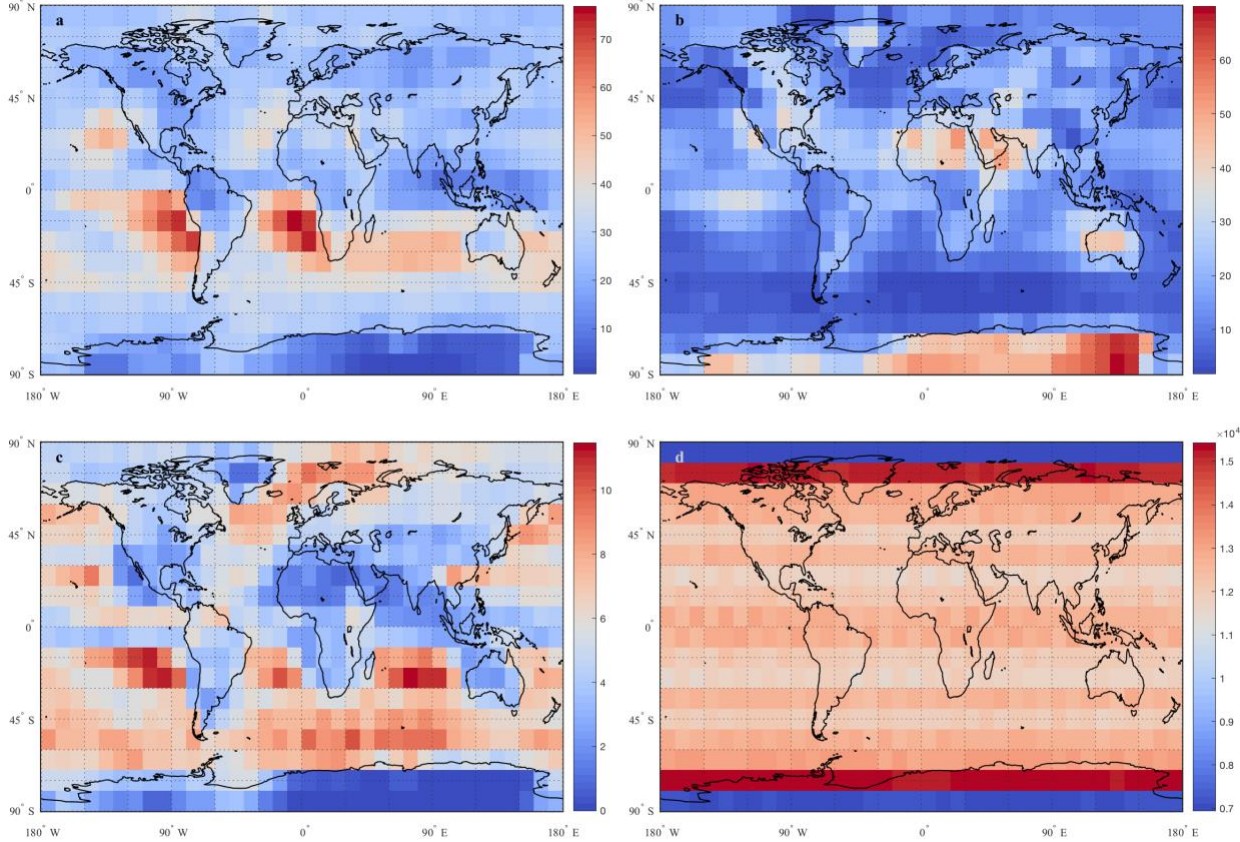

**Figure 2. Geographic distribution of cloud scenes: (Sc) on panel (a), (nc) on panel (b) and (Ac,Sc) on panel (c). Panel (d) show the geographic distribution of the sum of counts of all the (194) observed cloud scenes. The counts of the three other panels are normalized in each grid cell with the count of the corresponding grid cell in the lower right panel and are reported in percent.**



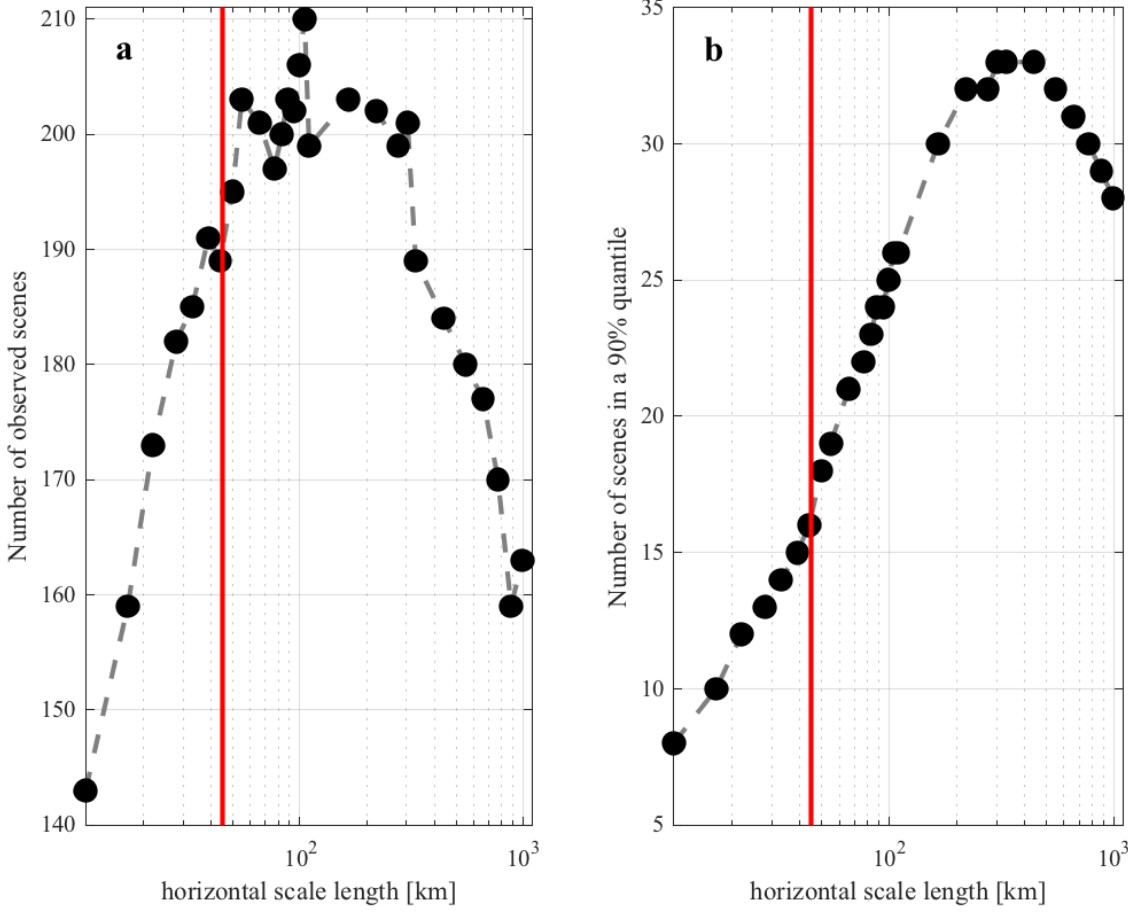

**Figure 3.** Number of observed scenes as a function of the horizontal scale length used to define the scene (a). Number of scenes observed in the 90[th] percentile as a function of horizontal scale length used to define the scene (b). The vertical red lines approximate the scale of the AIRS/AMSU FOR.





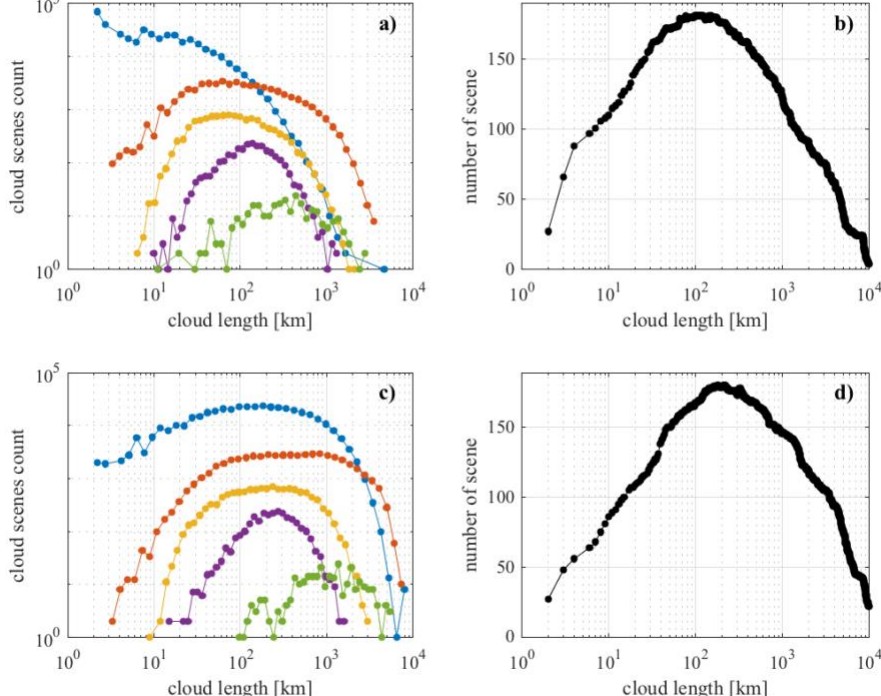

**Figure 4. Distribution of a) minimum and c) maximum length for 5 of the first 200 cloud. The five scenes are, from top to bottom: (AlCu,Sc) in blue, (AlSt,Sc,Cu) in orange, (ci,AlSt,Cu,DC) in yellow, (AlSt,AlCu,Ns,DC) in purple and (ci,AlSt,AlCu,St,Sc) in green and their respective ranks are 1, 26, 51, 76 and 101. On panels b) and d), the number of scenes were obtained by summing the number of scenes present a different lengths.**





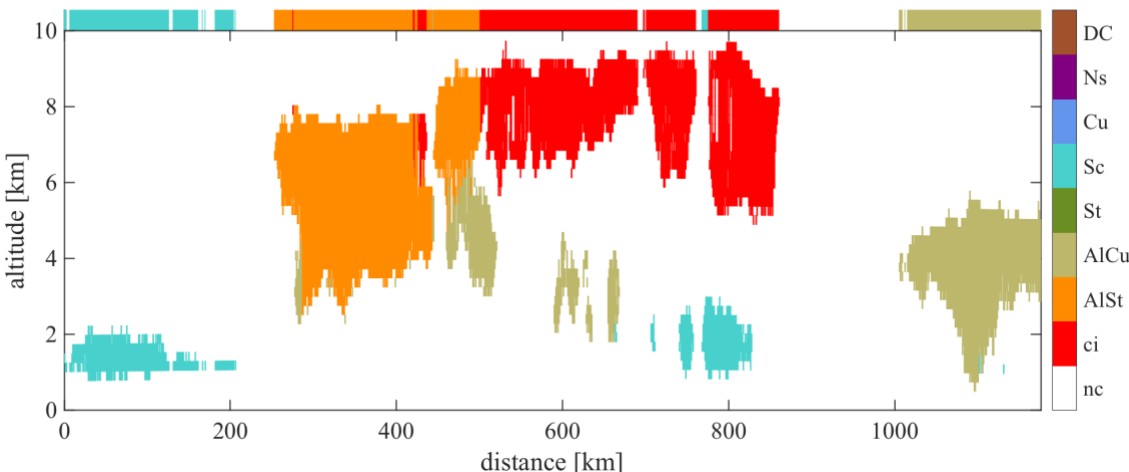

**Figure 5. Cloud profile defined by the values of the cloud_scenario variables of the 2b-CLDCLASS. Each color corresponds to a different cloud type. Color segments on top of the figure indicate the horizontal extent of a cloud measured at its top.**



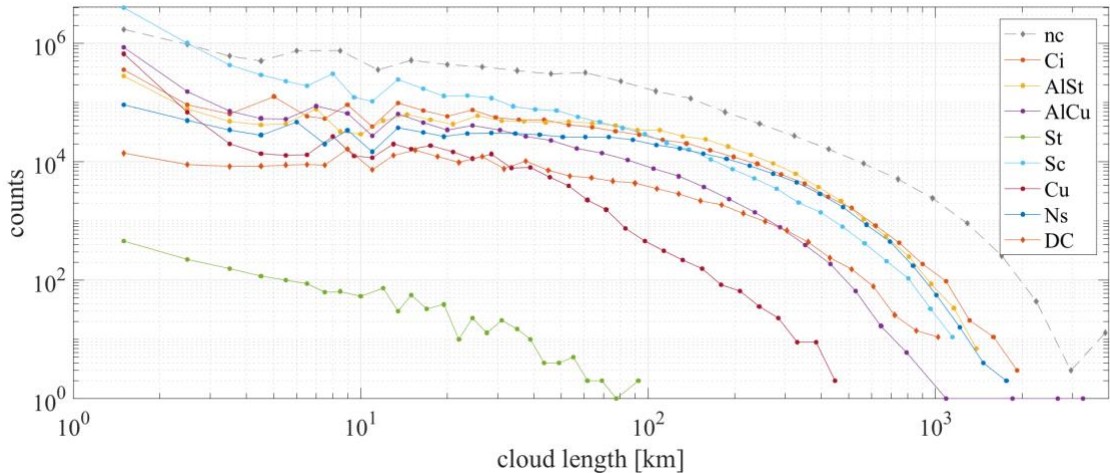

**Figure 6. Horizontal cloud chord length frequency histograms for each of the eight CloudSat cloud types and clear. The cloud chord length was obtained at the cloud top (see Fig. 5) unlike that obtained in Guillaume et al. (2018).**



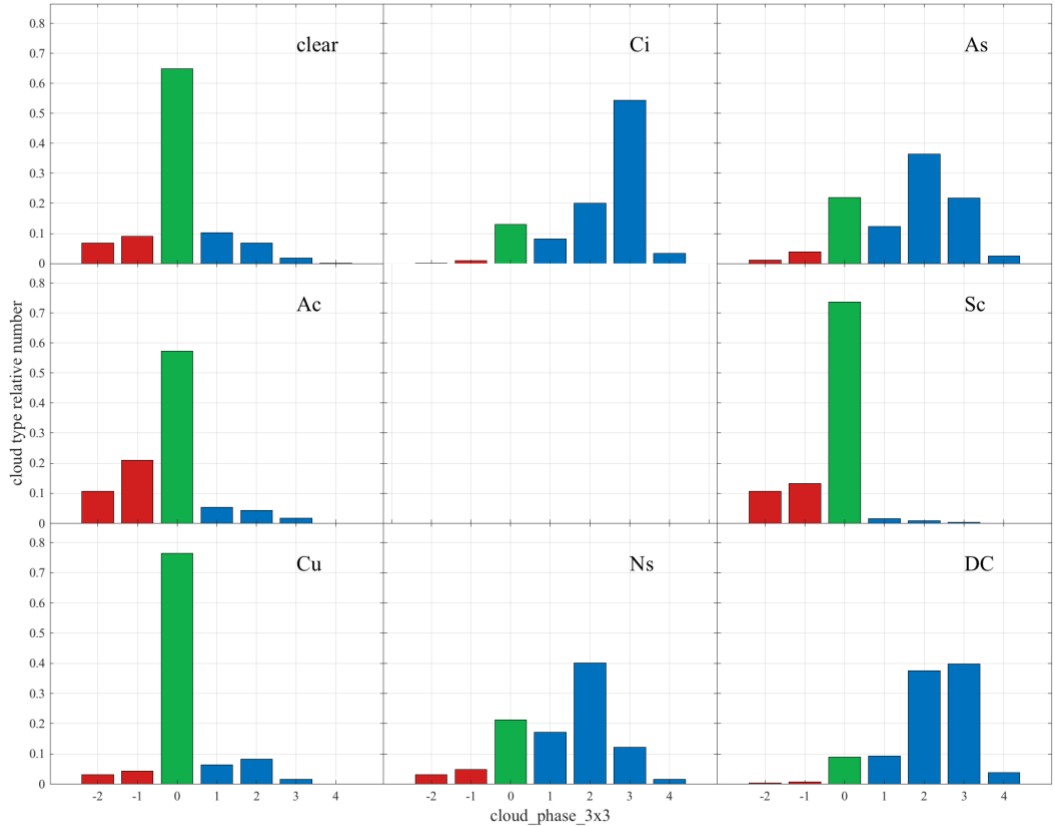

**Figure 7.** AIRS cloud_phase_3x3 variable for single cloud types (all CloudSat pixels within one AIRS footprint have the same cloud type but no clear scenes). The red, green, and blue bars indicate liquid, undetermined, and ice phase, respectively.





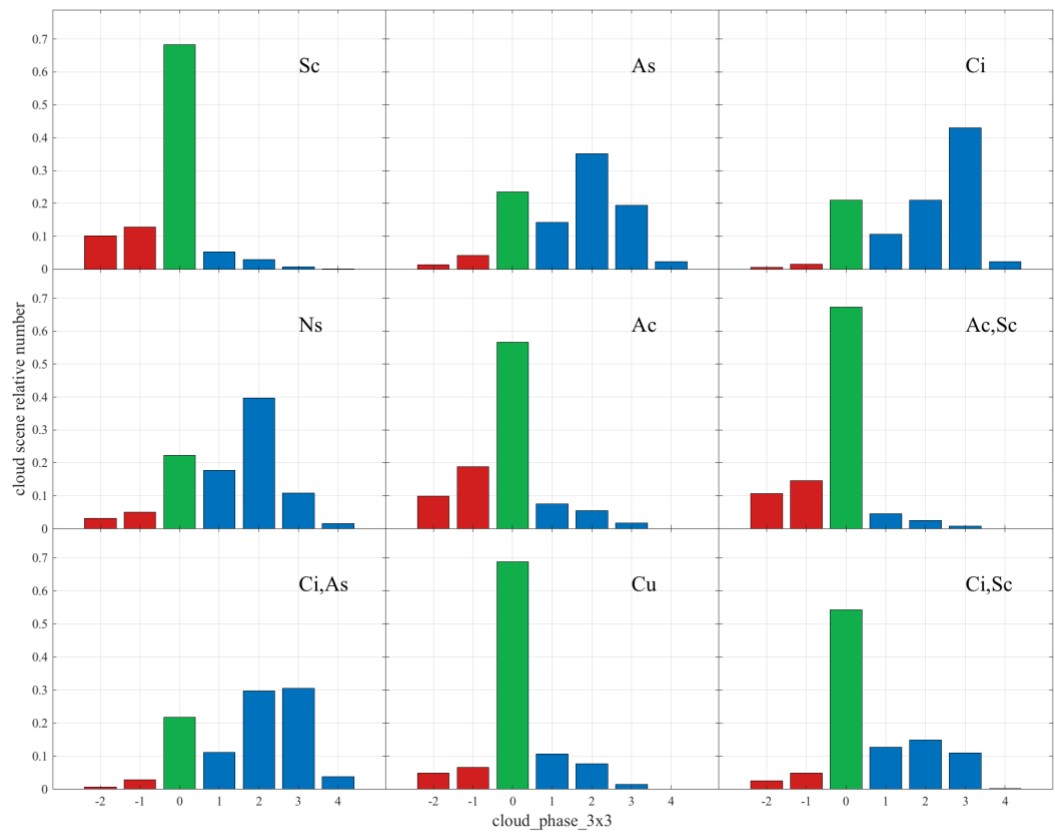

5   **Figure 8. AIRS cloud_phase_3x3 variable for the first mixed scenes of the 90th percentile. The first 'mixed' scene, with only clear fields, nc, was omitted for clarity and because it is the same as the nc cloud_phase_3x3 barplot of pure nc (see Fig. 7).**