# Peer review of "Footprint-scale cloud type mixtures and their impacts on Atmospheric Infrared Sounder cloud property retrievals"

_Atmospheric Measurement Techniques, 2018_

## Referee Comment (RC1) · Anonymous Referee #2 · 5 Jan 2019

In this paper AIRS measurements are combined with Cloudsat observations to obtain statistics of the variation of cloud types within the AIRS field of view. These are then used to study the variation of cloud properties per cloud scene and the sensitivity of ice cloud properties retrievals to the variation of cloud types within the pixels.

The study is interesting and the methods are generally sound. However, some issues with the presentation and the analysis need to be addressed before the paper can be accepted for publication. Below the major and minor comments are listed:

1) The Cloudsat cloud types are used. However, these type names are not very quantitative. Please indicate how the different types are defined and summarize how they

[Figure]

are derived from Cloudsat observations. The reader should not have to dig through other papers to be able to understand the data presented here. For instance, what are the altitude boundaries distinguishing low, middle and high clouds? It is also not clear to me how cumulus and stratus can be distinguished using a single radar profile.

2) Please be consistent with cloud type names throughout the paper. For example, sometimes "As" is used and sometimes "AlSt". Sometimes "clear sky" is used, sometimes "nc". Also, mixtures are denoted with commas, although clear-sky is left out in this notation. I suggest to indicate mixtures of cloud types and clear sky consistently, e.g., "As, nc". To give an example how this is confusing: Looking at Fig. 6, It was unclear how cumulus can have a length scale of 400 km, but I guess that's possible because clear-sky is mixed in. This can be made clearer by naming this mixture "Cu, nc" (This then excludes pure "Cu" cases. If these are included too, then I suggest using something like "Cu + Cu, ns").

3) Two different footprint sizes are considered, namely the AIRS/AMSU footprint of ∼45 km and the AIRS footprint of ∼15 km. It is a bit unclear throughout the paper which analysis is applied to which of the two footprints. In any case, the two different scales need to be addressed more consistently throughout the paper (in addition to the abstract). For example, Section 3.1 focuses on the AMSU footprint but not on the AIRS footprint. If I'm not mistaken, the AIRS cloud retrievals are performed on the AIRS ∼15 km footprint, so the last section focuses only on that scale (If so, state this clearly in the paper.) Please expand the discussion in section 3.1 to the AIRS footprint as well and add a figure similar to figure 1 for the AIRS footprint. Furthermore, I assume Fig. 2 is showing statistics for the AMSU footprint. If this is true, please state that in the paper. I suggest adding figures similar to Fig 2. for the AIRS footprint, or at least discussing the (lack of) differences between AIRS and AMSU global distributions. In fig. 3, 4 and 6, indicate the size of AIRS and AMSU pixels.

4) If my interpretation is correct, one goal is to compare AIRS cloud retrievals for cases with a single cloud type in the footprint with clear sky mixed in versus without clear

sky. However, a comparison is made between 1) cases with a single cloud type in the footprint and no clear sky and 2) cases with a single cloud type in the footprint either with and without clear sky. Thus, the case 2 set also contains the case 1 set in addition to the mixture of clear sky and a single cloud type. Therefore, the differences in properties listed in the paper are not representing the differences between cases without clear sky mixed in versus those with clear sky mixed in. From the information in the paper, we cannot deduce the relative number of cases per cloud type with versus without clear sky mixed in. If, for example, the number of cases without clear sky mixed in is much larger than those with clear sky, then the small differences shown between table 2 and 3 and figures 7 and 8 are surely expected. I suggest the following: 1) Include a table or figure showing the relative number of single type cases with and without clear sky mixed in. 2) For the single types listed in table 3 and figure 8 include only cases that also include clear sky. (Or add a table and figure showing this, leaving table 3 and fig. 8 as is.) To include a complete comparison between table 2 and 3 and figures 7 and 8, I suggest to also include the deep convection type (mixed with clear sky) in table 3 and figure 8. 3) Adjust the discussion of the differences accordingly.

Minor comments:

At the start of Page 11 it is stated that "the AIRS footprint ( $\sim$ 15 km ) is commensurate with the dimension of a single cloud so that the most frequent observations involve the characteristics of one cloud." What is meant by this statement? Many clouds, for example cumulus, have typical scales much smaller than that.

Table 1 and 2: Mean optical thicknesses are given here, but these are biased low because AIRS is not sensitive to any variation past an optical thickness of $\sim$5. Please make this clear in the text. The true mean optical thickness of most types will be larger than the AIRS-retrieved means shown here.

I think the caption of figure 4 is missing the word "scenes" at end of first sentence. It is noted in the text that Fig. 4b and 4d are similar to Fig. 3, but it is not directly clear what

the difference in the calculations are. Please explain.

I noted that the statistics listed in Table 3 are different than in Fig. 1, and this might be due to the different scales of AIRS vs AMSU. (See main comments.) However, for the ice properties to be retrieved, the phase needs to be identified as ice, so are these proportions for ice clouds only? Is that also explaining the difference in statistics compared to figure 1? Please explain in the text.
* * *

---

## Referee Comment (RC2) · Anonymous Referee #1 · 8 Jan 2019

The manuscript attempts to answer the following questions: How many different cloud types co-exist within a particular area? What cloud type mixtures are more prevalent? How do answers to the above two questions depend on area size? (side question that emerges: at what spatial scale does one encounter the greatest diversity of distinct cloud type mixtures?). These all sound kind of philosophical questions, but the authors find practical relevance (at least for the first question) for AIRS (and AMSU) scales cloud retrievals. The link to AIRS allows the authors to make one the major compromises of the study: only cloud type identification of the topmost cloudy layer matters because that's where AIRS is most sensitive even though the data source identifying cloud type provides vertical profile information. The other major compromise is

that when identifying cloud mixtures, the frequency of occurrence of each cloud type does not matter, in other words cloud mixtures consisting of the same cloud types are treated as equivalent even if the contributions of a cloud type are different. These two simplifications, along with an additional one where the spatial arrangement of the cloud types is ignored allow the authors to reduce the dimensionality of the problem and make the analysis tractable. This is overall quite a difficult paper to read, but I find the results of the first part quite fascinating (I was less excited about the implications for AIRS retrievals–although I understand that these findings are important for understanding the quality of the AIRS retrievals), so I recommend acceptance of the article to AMT. As you can see below, I have some inquiries some of which are also of the philosophical kind I'd like the authors to consider.

– What does the cloud type from 2B-CLDCLASS mean? The names of cloud types are the same as the ones used by surface observers, but are they related? Some description of the physical meaning of the cloud types given their method of identification by the 2B-CLDCLASS algorithm is needed. I'm sure the authors are aware that another version of the product currently exists, 2B-CLDCLASS-LIDAR where the CALIPSO lidar assists in the identification of the cloud type. Why was this newer product not used? (I suspect the authors may have started the work before this product was released). If the authors were to use 2B-CLDCLASS-LIDAR and the results changed in a major way, how would that undermine the fundamentals and motivation for the first part of the study? What if a completely different cloud type product was used, e.g., based on passive satellite observations where cloud type is identified by location in a cloud-top-pressure/cloud-optical-thickness joint histogram (the authors briefly touch on this in the last paragraph, but only with regard to the AIRS application – I'm more interested in the cloud scene climatology aspects)?

– It seems to me that the results depend completely on how frequently 2B-CLDCLASS identifies certain cloud types based on its internal definitions. Yes, the authors do not often find mixtures containing stratus (St) simply because St is extremely rare in

2B-CLDCLASS, probably unrealistically so given other methods identifying St (I mean, cloud types will always be loosely defined). I think one figure that the paper needs to include is the global frequency of the different cloud types according to 2B-CLDCLASS at its native resolution. This will give immediately clues on why certain cloud type mixtures (scenes) will be rare right off the bat (the authors kind of bring this this up already in some instances, e.g., p. 6, line 4). With DC, Cu, and St being rare according to 2B-CLDCLASS, one would expect that scenes containing those will also be rare.

– It is unfortunate that the abbreviation for certain cloud types changes throughout the text, tables, and figures: As becomes AlSt, Ac becomes AlCu, Ci becomes ci, DC becomes Dc, and so forth. Please fix and make consistent throughout!

– I don't understand panel d in Fig. 2. Whatever it depicts, it does not appear to have a very interesting pattern!

– I recognize that the authors make a valiant effort in section 3.3, but that part of the paper remains a hard read. In this section, line 8 of p. 8 indicates that 200 possible mixed scenes were identified which seems to contradict the 194 figure quoted earlier (p. 5, line 23). Are these numbers for areas of different size (e.g., a third figure of 210 different scenes emerges for 105 km). Please clarify, 194, 200, and 210. Moreover, I found odd that the authors state (p. 7 line 2) that "The maximum number of observed cloud scenes (210) at a particular horizontal scale (105 km) remains unexplained" when the section that immediately follows tries to explain exactly that. Am I missing the subtle distinction? Section 3.3 tries to explain why the maximum number happens at 105 (not sure it succeeds), but why this maximum number is 210 remains as the unexplained mystery?

– Can the same scale be used for the y-axis of Figs. 7 and 8? You say that that the common panels of these two figures (single cloud type scenes) should look very similar (inclusion of clear-sky notwithstanding), but the comparison is hampered by different y-axis range.

– Somewhere in section 2 mention what the maximum optical thickness retrievable by AIRS is.

---

## Author Comment (AC1) · 26 Apr 2019

Reviewer #1

We thank the reviewer for the insightful comments (**bold font**) and we have replied to each of the comments and queries below each comment (regular font) and modifications to the manuscript are in *italic font*.

**The manuscript attempts to answer the following questions: How many different cloud types co-exist within a particular area? What cloud type mixtures are more prevalent? How do answers to the above two questions depend on area size? (side question that emerges: at what spatial scale does one encounter the greatest diversity of distinct cloud type mixtures?). These all sound kind of philosophical questions, but the authors find practical relevance (at least for the first question) for AIRS (and AMSU) scales cloud retrievals. The link to AIRS allows the authors to make one the major compromises of the study: only cloud type identification of the topmost cloudy layer matters because that's where AIRS is most sensitive even though the data source identifying cloud type provides vertical profile information. The other major compromise is that when identifying cloud mixtures, the frequency of occurrence of each cloud type does not matter, in other words cloud mixtures consisting of the same cloud types are treated as equivalent even if the contributions of a cloud type are different. These two simplifications, along with an additional one where the spatial arrangement of the cloud types is ignored allow the authors to reduce the dimensionality of the problem and make the analysis tractable.**

We appreciate the reviewer's synthesis of the paper and agree these are the major thrusts of the first aspect of the paper.

**This is overall quite a difficult paper to read, but I find the results of the first part quite fascinating (I was less excited about the implications for AIRS retrievals–although I understand that these findings are important for understanding the quality of the AIRS retrievals), so I recommend acceptance of the article to AMT. As you can see below, I have some inquiries some of which are also of the philosophical kind I'd like the authors to consider.**

We have made some edits and changes that follow from the reviewer suggestions that we hope make the manuscript more readable. We do appreciate the reviewer's point of view regarding how this paper may be viewed in two distinct pieces. However, our end goal of this work was to show practical relevance to the cloud scene variability and ultimately establish why it is important to describe cloud scenes on a pixel-by pixel basis. The reasons are hopefully clearer in the revised version as we have more carefully documented AIRS retrievals of cloud phase and ice cloud properties as a function of whether the scene contains one cloud type or multiple cloud types, or whether the scene is completely or partly cloudy, instead of reporting them in a slightly convoluted manner. These changes are closely coupled to those arising from reviewer #2's comments (please refer to replies to reviewer #2 for further detail).

**– What does the cloud type from 2B-CLDCLASS mean? The names of cloud types are the same as the ones used by surface observers, but are they related? Some description of the physical meaning of the cloud types given their method of identification by the 2B-CLDCLASS algorithm is needed.**

The 2B-CLDCLASS algorithm is described in Sassen and Wang (2005, 2008) that is referenced in the manuscript. As stated in Sassen and Wang (2008), the algorithm is based on earlier work by the same authors and combines the measurements of ground-based multiple remote sensors. They report having tested the results against surface observer cloud reports. We have included some additional text at the beginning of Section 2.2 to clarify:

*"The CloudSat 2B-CLDCLASS product is used in this work and the algorithm is described in Sassen and Wang (2005, 2008). As summarized in Sassen and Wang (2008) and previous works, the algorithm uses methods developed from ground-based multiple remote sensors that have been tested against surface observer-based cloud typing reports. The cloud classification occurs in two steps. First, a clustering analysis is performed to group cloud profiles into cloud clusters. Secondly, classification methods are used to classify clouds into different cloud types. The decision trees guiding the classification are complex and are based on 23 variables derived from the clustering analysis of the first stage. Geometric quantities such as cloud base, top, and horizontal extents are present in decision trees (Sassen and Wang, 2005)."*

**I'm sure the authors are aware that another version of the product currently exists, 2B-CLDLASS-LIDAR where the CALIPSO lidar assists in the identification of the cloud type. Why was this newer product not used? (I suspect the authors may have started the work before this product was released). If the authors were to use 2B-CLDCLASS-LIDAR and the results changed in a major way, how would that undermine the fundamentals and motivation for the first part of the study?**

This is a good question and a fair one to ask. We are motivated by the results of Kahn et al. (2018) that suggests larger particle sizes in convective clouds compared to thin cirrus. The 2B-CLDCLASS product is better suited for differentiating cloud types other than small particle thin cirrus, in which 2B-CLDCLASS-LIDAR would excel. While AIRS is very sensitive to thin cirrus, the sensitivity of AIRS ice cloud particle size most strongly responds to clouds around an optical thickness on the order of 1.0. If we used 2B-CLDCLASS-LIDAR, the statistics would be weighted to the detection of vast areas of thin cirrus in layers above optically thicker clouds and elsewhere in absence of other cloud types. The Ci classification dominates in the 2B-CLDCLASS-LIDAR data set and will blur the signals of cumulus and deep convective cloud types capped by thin cirrus. As we describe in the paper, the simplifications required to make the approach tractable require us to quantify cloud type at cloud top, and thus we would lose the discriminatory ability for clouds that occur just below a thin layer of cirrus. We expect that the mixtures would be profoundly different in 2B-CLDCLASS-LIDAR, with much less ability to demonstrate AIRS' skill at obtaining larger particle sizes at convective cloud tops (Kahn et al. 2018). Ultimately, we argue that 2B-CLDCLASS is a more appropriate tool for this work. We have included some additional text in Section 2.2 to clarify:

*"Lastly, the results of Kahn et al. (2018) suggest larger ice cloud particle sizes occur at convective cloud tops compared to thin cirrus at the same cloud top temperature. Given the key assumption of cloud typing only at cloud top, the 2B-CLDCLASS product is better suited for identifying convective clouds in AIRS apart from stratiform clouds, the latter of which are dominant in 2B-CLDCLASS-LIDAR. If 2B-CLDCLASS-LIDAR was used, the statistics would be weighted towards the detection of vast areas of cirrus in thin layers above and in proximity to convective clouds. The Ci classification dominates in 2B-CLDCLASS-LIDAR at cloud top and will blur the signals of underlying cumulus and deep convective cloud types that are capped by thin cirrus."*

**What if a completely different cloud type product was used, e.g., based on passive satellite observations where cloud type is identified by location in a cloud-top- pressure/cloud-optical-thickness joint histogram (the authors briefly touch on this in the last paragraph, but only with regard to the AIRS application – I'm more interested in the cloud scene climatology aspects)?**

As the reviewer certainly knows, how one goes about "typing clouds" is not a settled research topic and is sensitive to the instrument sampling, radiance sensitivity, wavelength, underlying assumptions, and so forth. In the conclusions, we touch on the results of Wang et al. (2016) where comparisons of CloudSat cloud types as used here in this work are compared to ISCCP-like categories derived from the MODIS imager, which could be used in place of the CloudSat cloud typing. We expect that the CloudSat radar will have more skill in discriminating convective clouds from stratiform clouds than passive sensors, as these two types of clouds show strong differences in the AIRS microphysical retrievals. For many cloud types, the detection is similar between passive and active; please see Wang et al. (2016) for specifics.

**– It seems to me that the results depend completely on how frequently 2B-CLDCLASS identifies certain cloud types based on its internal definitions. Yes, the authors do not often find mixtures containing stratus (St) simply because St is extremely rare in 2B-CLDCLASS, probably unrealistically so given other methods identifying St (I mean, cloud types will always be loosely defined).**

We agree.  Please refer above for our generalized perspective. It is well established that 2B-CLDCLASS contains very little stratus because of ground clutter in the bottom 3-4 bins.

**I think one figure that the paper needs to include is the global frequency of the different cloud types according to 2B-CLDCLASS at its native resolution. This will give immediately clues on why certain cloud type mixtures (scenes) will be rare right off the bat (the authors kind of bring this this up already in some instances, e.g., p. 6, line 4). With DC, Cu, and St being rare according to 2B-CLDCLASS, one would expect that scenes containing those will also be rare.**

With regard to plan view maps and zonal averaged plots of cloud type frequencies from 2B-CLDCLASS, these have been reported in the literature, in particular we are referring to Sassen

and Wang (2008), their Figure 1 for the plan view and their Figure 2 for the zonal average. We do not see a need to repeat these results in this paper. With regard to the relative histogram counts of cloud type as a function of length scale, these are depicted in Figure 6 in the manuscript. The percentages of cloud type and cloud scene frequencies are also reported in Tables 3-6 in the revised manuscript.

We have added the following statement in Section 2.2 to refer the reader for more detail on cloud type frequencies: "*Plan view and zonal average frequencies of 2B-CLDCLASS cloud types at its native resolution are reported in Sassen and Wang (2008).*"

**It is unfortunate that the abbreviation for certain cloud types changes throughout the text, tables, and figures: As becomes AlSt, Ac becomes AlCu, Ci becomes ci, DC becomes Dc, and so forth. Please fix and make consistent throughout!**

We have checked the notation throughout the revised manuscript and fixed any inconsistencies in the labeling. We intended to follow the notation cumulus (Cu), stratocumulus (Sc), stratus (St), altocumulus (Ac), altostratus (As), nimbostratus (Ns), cirrus (Ci), deep convective (Dc) clouds and a ninth classification of clear sky designated no cloud (nc). In addition to text changes, we corrected figure 1, 5, 6 and 7.

**I don't understand panel d in Fig. 2. Whatever it depicts, it does not appear to have a very interesting pattern!**

We agree that these are confusing and have removed panels b and d in figure 2 as they are not key pieces of information for the manuscript.

**I recognize that the authors make a valiant effort in section 3.3, but that part of the paper remains a hard read. In this section, line 8 of p. 8 indicates that 200 possible mixed scenes were identified which seems to contradict the 194 figure quoted earlier (p. 5, line 23). Are these numbers for areas of different size (e.g., a third figure of 210 different scenes emerges for 105 km). Please clarify, 194, 200, and 210. Moreover, I found odd that the authors state (p. 7 line 2) that "The maximum number of observed cloud scenes (210) at a particular horizontal scale (105 km) remains unexplained" when the section that immediately follows tries to explain exactly that. Am I missing the subtle distinction? Section 3.3 tries to explain why the maximum number happens at 105 (not sure it succeeds), but why this maximum number is 210 remains as the unexplained mystery?**

In Section 3.1, we describe the statistics of cloud scenes observed at the AMSU resolution at 45-km scales. There are 194 observed cloud scenes at this resolution. In section 3.2, we quantify the scale dependence of cloud scene statistics. The number of cloud scenes is shown to first increase with resolution then decrease as scales get large. The maximum number of cloud scenes is 210 and is observed at a resolution of 105 km. Both of the numbers 194 and 210 appear as 2 points on figure 3a. The former is indicated with the intersection of the red line and the curve, and the latter is just the value at the maximum.

The third number mentioned by the reviewer is 200 and it is a result of the procedure described in section 3.3. We shortened the name of this section 3.3 "Generalizing to all scales" as we understood from the reviewer's comment that the title was potentially confusing or misleading. The number of 200 mixed scenes is identified *independently of a grid resolution*.

Therefore, there are 194 cloud scenes observed at a resolution of ~45 km, 210 scenes observed at a resolution of 105 km and 200 scenes identified with a procedure independent from a regular grid.

We have added the following text at the start of Section 3.2 to clarify: "*In Section 3.1, the relative frequencies of cloud scenes were derived for exact matches of AIRS and AMSU observations to the CloudSat ground track. As the CloudSat ground track can oscillate over several AIRS FOVs across a scan line within a given orbit, the numbers of coincident CloudSat profiles matching to AIRS and AMSU will vary. Below, cloud scenes are derived independently of the specific AIRS and AMSU collocation geometry.*"

As far as the quote mentioned by the reviewer on p. 7 line 2, that was intended to be a transition from Section 3.2 to 3.3 and as motivation for why we calculated the scale dependence of cloud scenes independently of a particular grid resolution. We have changed the text to the following to clarify: "*The reasons for the maximum number of observed cloud scenes (210) at a particular horizontal scale (105 km) are not immediately clear.*" Then we have changed the text on p.7 lines 4-5 to further clarify: "*A simple conceptual model is described below that is able to approximate the results of Fig. 3 and offers some insight for the observed maximum frequency of cloud scenes and the spatial scale at which it occurs.*"

We have rewritten some of the paragraph on p. 8 lines 11-20 as follows: "*A total of 200 out of 247 possible mixed scenes were identified. The minimum and maximum length occurrence frequencies of five cloud scenes (Ac,Sc), (As,Sc,Cu), (Ci,As,Cu,dc), (As,Ac,Ns,dc) and (Ci,As,Ac,St,Sc) selected randomly from the 200 present in the two-year record, are shown in Fig. 4a and 4c, respectively. Recall that the maximum length is defined from relation (2), while the minimum length is defined from relation (3), with an illustrative example shown for (Ac,Sc). From top to bottom, their respective ranks are 1, 26, 51, 76 and 101. It is striking that each frequency histogram in Fig. 4a and 4c is not monotonic and displays a frequency maximum between 100 and 1000 km. Consequently, the sum of all (200) observed mixed scenes across length scales will result in a curve with a maximum and these are shown in Fig. 4b and 4d. Both curves are very similar to Fig. 3a and have maxima for about 180 observed scenes at 77 km and 174 km, respectively. Using the methodology outlined in (1) to (3), the scale dependence of the number of observed scenes shows that the maximum will occur somewhere between 77 and 174 km.*"

**– Can the same scale be used for the y-axis of Figs. 7 and 8? You say that that the common panels of these two figures (single cloud type scenes) should look very similar (inclusion of clear-sky notwithstanding), but the comparison is hampered by different y-axis range.**

We regenerated figures 7 and 8 with the same ordinate scales. However, the reviewer should keep in mind that each bin in Figures 7-10 (revised) are normalized by the total number of cloud scenes (within an AIRS footprint) for single or multiple cloud types, or partly cloudy or fully cloudy scenes. We have added the following text to the caption of Figure 7: "*Each histogram sums to 1.0 and does not show the numbers of counts relative to another histogram. Relative counts could be inferred from the percentages listed in the 2$^{nd}$ to left column of Table 3.*"

**– Somewhere in section 2 mention what the maximum optical thickness retrievable by AIRS is.**

We added the following sentence on line 12 page 5: "*The AIRS sampling includes nearly all ice clouds with τi > 0.1, while the maximum values of τi asymptote to values near 6–8 (e.g. Kahn et al., 2015)*."

---

## Author Comment (AC2) · 26 Apr 2019

Reviewer #2

We thank the reviewer for the insightful comments (**bold font**) and we have replied to each of the comments and queries below each comment (regular font) and modifications to the manuscript are in *italic font*.

**In this paper AIRS measurements are combined with Cloudsat observations to obtain statistics of the variation of cloud types within the AIRS field of view. These are then used to study the variation of cloud properties per cloud scene and the sensitivity of ice cloud properties retrievals to the variation of cloud types within the pixels. The study is interesting and the methods are generally sound. However, some issues with the presentation and the analysis need to be addressed before the paper can be accepted for publication. Below the major and minor comments are listed:**
**1) The Cloudsat cloud types are used. However, these type names are not very quantitative. Please indicate how the different types are defined and summarize how they are derived from Cloudsat observations. The reader should not have to dig through other papers to be able to understand the data presented here. For instance, what are the altitude boundaries distinguishing low, middle and high clouds? It is also not clear to me how cumulus and stratus can be distinguished using a single radar profile.**

According to Sassen and Wang (2005), the cloud classification occurs in two steps. First, a clustering analysis is performed to group cloud profiles into cloud clusters. Secondly, classification methods are used to classify clouds into different cloud types. The decision trees guiding the classification are complex and based on 23 variables derived from the clustering analysis of the first stage. Conditions upon geometric quantities such as cloud base, top, and horizontal extents are present in decision trees (page 32 to 35 in Sassen and Wang, 2005).

One of our responses to reviewer #1 is relevant here and we added text to the manuscript (repeated from response to reviewer #1): "*The CloudSat 2B-CLDCLASS product is used in this work and the algorithm is described in Sassen and Wang (2005, 2008). As summarized in Sassen and Wang (2008) and previous works, the algorithm uses methods developed from ground-based multiple remote sensors that have been tested against surface observer-based cloud typing reports. The cloud classification occurs in two steps. First, a clustering analysis is performed to group cloud profiles into cloud clusters. Secondly, classification methods are used to classify clouds into different cloud types. The decision trees guiding the classification are complex and are based on 23 variables derived from the clustering analysis of the first stage. Conditions upon geometric quantities such as cloud base, top, and horizontal extents are present in decision trees (Sassen and Wang, 2005).*"

In the conclusions, we describe the results of Wang et al. (2016) where comparisons of CloudSat cloud types as used here in this work are compared to ISCCP-like categories derived from the MODIS imager, which could be used in place of the CloudSat cloud typing. For many cloud types, the detection is similar between passive and active; please see Wang et al. (2016) for specifics.

**2) Please be consistent with cloud type names throughout the paper. For example, sometimes "As" is used and sometimes "AlSt". Sometimes "clear sky" is used, sometimes "nc". Also, mixtures are denoted with commas, although clear-sky is left out in this notation. I suggest to indicate mixtures of cloud types and clear sky consistently, e.g., "As, nc". To give an example how this is confusing: Looking at Fig. 6, It was unclear how cumulus can have a length scale of 400 km, but I guess that's possible because clear-sky is mixed in. This can be made clearer by naming this mixture "Cu, nc" (This then excludes pure "Cu" cases. If these are included too, then I suggest using something like "Cu + Cu, ns").**

We have checked the notation throughout the revised manuscript and fixed any inconsistencies in the labeling. We intended to follow the notation cumulus (Cu), stratocumulus (Sc), stratus (St), altocumulus (Ac), altostratus (As), nimbostratus (Ns), cirrus (Ci), deep convective (Dc) clouds and a ninth classification of clear sky designated no cloud (nc). In addition to text changes, we corrected figure 1, 5, 6 and 7.

Cumulus lengths reported in Figure 6 are for pure Cu and not a blend of clear and Cu (Cu,nc). Cumulus clouds of any length are possible according to the decision tree of low cloud classifier (page 34 of Sassen and Wang, 2005) when a cluster cloud fraction is smaller than 0.25. Sassen and Wang (2005) do not indicate that a maximum cluster length is prescribed, so long cumulus clouds might result from the unlikely situation where a long cumulus cloud is part of a cluster with cloud fraction less than 0.25.

The fact is that we observed long clouds in each cloud category of the 2B-CLDCLASS CloudSat product. One may quibble about this categorization but the algorithm most closely associates that cloud with Cu. For example, we found that the longest cumulus cloud is located in granule 03498 of the 2B-CLDCLASS product, 2006358131321_03498_CS_2B-CLDCLASS_GRANULE_P_R04_E02.hdf using the cloud type defined at the cloud top as described in the manuscript. The very long Cu cloud starts at profile # 25681 and ends at profile # 26118. It therefore has length of 438 CloudSat profiles or about 482 km. The location of this long cumulus cloud is shown in the figure below:

[Figure]

This fact emphasizes that no classification is perfect, that some misclassification is inevitable and that some cloud physical states are ambiguous for a given classification scheme. We used the 2B-CLDCLASS CloudSat product for our study but a comprehensive critical analysis of this product is beyond the scope of our paper.

**3) Two different footprint sizes are considered, namely the AIRS/AMSU footprint of ~45 km and the AIRS footprint of ~15 km. It is a bit unclear throughout the paper which analysis is applied to which of the two footprints. In any case, the two different scales need to be addressed more consistently throughout the paper (in addition to the abstract). For example, Section 3.1 focuses on the AMSU footprint but not on the AIRS footprint. If I'm not mistaken, the AIRS cloud retrievals are performed on the AIRS ~15 km footprint, so the last section focuses only on that scale (If so, state this clearly in the paper.) Please expand the discussion in section 3.1 to the AIRS footprint as well and add a figure similar to figure 1 for the AIRS footprint. Furthermore, I assume Fig. 2 is showing statistics for the AMSU footprint. If this is true, please state that in the paper. I suggest adding figures similar to Fig 2. for the AIRS footprint, or at least discussing the (lack of) differences between AIRS and AMSU global distributions. In fig. 3, 4 and 6, indicate the size of AIRS and AMSU pixels.**

We have added a second panel to Fig. 1 for the AIRS FOV scale and have labeled each panel clearly with the two scales depicted. Note that the number of scenes that explains >90% of the observed scenes is different between the two scales.

We describe in section 2 that the analysis is first conducted at the AMSU FOR spatial scale and that this scale is 45 km. We replaced the initial notation "AIRS/AMSU FOR" with "AMSU FOR" throughout the manuscript to avoid confusion between the two different resolutions and their corresponding scales. After discussing scene observations at this scale in section 3.1, we extend the analysis as a function of resolution in sections 3.2 and 3.3. After showing the cloud scene resolution dependence in Section 3, we now describe the dependence of AIRS cloud properties on the different types of cloud scene in section 4. We added a sentence at the beginning of section 4, to emphasize that the resolution used for cloud scenes determination, in this section is AIRS, i.e. about 15 km: "*In this section, the scenes are determined at the AIRS FOV resolution (approximately 15 km).*"

We agree with the referee and added a sentence in the caption of figure 2 to specify the resolution scale: "*These scenes were observed at the AMSU FOV resolution (~45 km).*" We note that the AIRS version of Figure 2 is nearly identical and are not included. We have added the following text to the Fig. 2 caption: "*Similar plots of the AIRS FOV resolution (~15 km) are nearly identical (not shown).*"

We think it is a good idea to indicate the AIRS and AMSU resolutions in figures 3 and 6. New versions of these figures with vertical lines indicating both resolutions are added to the revised manuscript.

**4) If my interpretation is correct, one goal is to compare AIRS cloud retrievals for cases with a single cloud type in the footprint with clear sky mixed in versus without clear sky. However, a comparison is made between 1) cases with a single cloud type in the footprint and no clear sky and 2) cases with a single cloud type in the footprint either with and without clear sky. Thus, the case 2 set also contains the case 1 set in addition to the mixture of clear sky and a single cloud type. Therefore, the differences in properties listed in the paper are not representing the differences between cases without clear sky mixed in versus those with clear sky mixed in. From the information in the paper, we cannot deduce the relative number of cases per cloud type with versus without clear sky mixed in. If, for example, the number of cases without clear sky mixed in is much larger than those with clear sky, then the small differences shown between table 2 and 3 and figures 7 and 8 are surely expected. I suggest the following: 1) Include a table or figure showing the relative number of single type cases with and without clear sky mixed in. 2) For the single types listed in table 3 and figure 8 include only cases that also include clear sky. (Or add a table and figure showing this, leaving table 3 and fig. 8 as is.) To include a complete comparison between table 2 and 3 and figures 7 and 8, I suggest to also include the deep convection type (mixed with clear sky) in table 3 and figure 8. 3) Adjust the discussion of the differences accordingly.**

These are very helpful comments by the reviewer and we have made some significant changes in the manuscript to make the categorization of cloud scenes, and the differences in the cloud phase and ice microphysical retrievals clearer.

First, we have added a new Table 2 and a new (brief) Section 4.1 that summarizes the new breakdown in cloud scenes: "*Table 2 summarizes five types of scenes at the 15-km AIRS FOV scale: (i) clear sky, (ii) cloudy sky with one cloud type, (iii) partly cloudy sky with one cloud type, (iv), cloudy sky with multiple cloud types, and (v) partly cloudy sky with multiple cloud types. The raw counts and the relative percentages for the two-year observing period are shown. The dominance of clear sky (~31%) at 15-km is apparent and is consistent with an absence of thin cloud features in the 2B-CLDCLASS data set. Cloudy sky cloud scenes amount to less than 1% of all observed cloud scenes, with a vast majority as partly cloudy, and a majority of those with only one cloud type. This value is higher than that reported by Yue et al. (2011) for which the 45-km AMSU scale was used but is consistent with the factor of 3 difference in scale.*"

Second, we have reworked the original Tables 2 and 3 into four different tables (3-6) that show the ice cloud property statistics for the scene classification (2), (3), (4) and (5) listed above. In the case of (1), we have included clear sky in (2).

Third, we have similarly modified Figures 7 and 8 in the original manuscript to Figures 7-10 in the revised manuscript. They follow exactly the same scene classification divisions illustrated in Table 2.

We have added text in the manuscript that describe these new tables and figures and more clearly delineate the performance of cloud phase and ice cloud property retrievals by whether the cloud scene is completely versus partly cloudy, and whether the cloud scene contains a

single cloud type or multiple cloud types. We refer the reviewer to the revised version of the manuscript to see every change made, but the most substantial for Section 4.2 and 4.3 follow:

The first paragraph of Section 4.2 that addresses cloudy sky with one cloud type now reads as: "*The occurrence frequency histogram of the sum of all thermodynamic phase tests is shown for cloudy sky with one cloud type in Fig. 7. While these clouds account for a small percentage of the total number of AIRS FOVs, homogeneous cloud scenes serve as an ideal point of reference for establishing cloud phase sensitivity benchmarks. Overall, there is strong differentiation in the cloud thermodynamic phase among cloud scenes with single cloud types. Ice tests dominate Ci, Ns, Dc, and As, while liquid and undetermined tests dominate Ac, Sc, and Cu.*"

[revised manuscript text omitted]

**Minor comments:**
**At the start of Page 11 it is stated that "the AIRS footprint ( ∼ 15 km ) is commensurate with the dimension of a single cloud so that the most frequent observations involve the characteristics of one cloud." What is meant by this statement? Many clouds, for example cumulus, have typical scales much smaller than that.**

We agree that this statement could be confusing and we have removed the following from the revised manuscript:  "*These differences between single and mixed cloud scenes can be explained by the fact that the AIRS footprint (~ 15 km) is commensurate with the dimension of a single cloud so that the most frequent observations involve the characteristics of one cloud. Cloud length statistics, such as median lengths and median absolute deviations calculated from the data shown in Fig. 6, corroborate this statement.*"

**Table 1 and 2: Mean optical thicknesses are given here, but these are biased low because AIRS is not sensitive to any variation past an optical thickness of ~5. Please make this clear in the text. The true mean optical thickness of most types will be larger than the AIRS-retrieved means shown here.**

We added the following sentence line 12 of page 5: "*The AIRS retrieval sample includes nearly all ice clouds with $\tau_i > 0.1$, while the maximum values of $\tau_i$ asymptote to values around 6-8 (e.g. Kahn et al., 2015).*"

We discovered an error in Table 1, which are now fixed, and now the values on the third row actually report the median absolute deviation.

**I think the caption of figure 4 is missing the word "scenes" at end of first sentence. It is noted in the text that Fig. 4b and 4d are similar to Fig. 3, but it is not directly clear what the difference in the calculations are. Please explain.**

We added the word "*scene*" at the end of the first sentence of the figure 4 caption. Figure 3 shows the number of scenes observed at different FOR scale whereas Figure 4b and 4d represents the number of scenes as a function of a cloud scale (not as a function of FOR scale).

**I noted that the statistics listed in Table 3 are different than in Fig. 1, and this might be due to the different scales of AIRS vs AMSU. (See main comments.) However, for the ice properties to be retrieved, the phase needs to be identified as ice, so are these proportions for ice clouds only? Is that also explaining the difference in statistics compared to figure 1? Please explain in the text.**

The referee is correct in that these are only for ice clouds (the blue bars in Figures 7-10). The statistics in Table 3 are obtained at the AIRS FOR whereas statistics in Figure 1 were obtained at the AMSU FOR.